# Multiple Token Divergence: Measuring and Steering In-Context Computation Density

**Vincent Herrmann**[1,2*]**, Eric Alcaide**[1*]**, Michael Wand**[1] **& Jürgen Schmidhuber**[1,2]
[1] The Swiss AI Lab IDSIA/USI/SUPSI
   Lugano, Switzerland
[2] King Abdullah University of Science and Technology
   Thuwal, Saudi Arabia
[*] Equal contribution
{vincent.herrmann,eric.alcaide,michael.wand,juergen}@idsia.ch

## Abstract

Measuring the in-context computational effort of language models is a key challenge, as metrics like next-token loss fail to capture reasoning complexity. Prior methods based on latent state compressibility can be invasive and unstable. We propose Multiple Token Divergence (MTD), a simple measure of computational effort defined as the KL divergence between a model's full output distribution and that of a shallow, auxiliary prediction head. MTD can be computed directly from pre-trained models with multiple prediction heads, requiring no additional training. Building on this, we introduce Divergence Steering, a novel decoding method to control the computational character of generated text. We empirically show that MTD is more effective than prior methods at distinguishing complex tasks from simple ones. On mathematical reasoning benchmarks, MTD correlates positively with problem difficulty. Lower MTD is associated with more accurate reasoning. MTD provides a practical, lightweight tool for analyzing and steering the computational dynamics of language models.

## 1 Introduction

To solve unfamiliar and challenging problems, language models must perform sophisticated in-context computation (Brown et al., 2020; Lewkowycz et al., 2022). Can we tell whether, and to what extent, a model is making use of its computational capacity at any given moment? It is well-established that the next-token prediction loss offers little insight (Schmidhuber, 2010; Burda et al., 2018), as any particular reduction in loss can, in principle, be arbitrarily difficult to achieve (Bennett, 1988). A more promising approach is to quantify meaningful computation by measuring the entropy, or incompressibility, of a model's latent representations (Skean et al., 2025; Herrmann et al., 2025). This concept is rooted in the minimum description length (MDL) principle (Wallace and Boulton, 1968; Rissanen, 1978; Solomonoff, 1964; Vitányi, 2006; Elmoznino et al., 2024): if the most compact description of a sequence's structure, given the training data, is still long, then predicting that sequence is demanding due to a large search space. In other words, a task is "difficult" or "complex" if the shortest description of the solution program, given the training data, is long. If it is short, on the other hand, we consider the task "easy" and hence "boring". Unfortunately, in general, the length of the shortest description is uncomputable (Li et al., 2008). To make it tractable, the Prediction of Hidden States (PHi) loss was proposed as a measure of in-context computational complexity, quantifying the per-token information gain in a model's latent space (Herrmann et al., 2025). This work itself is based on the earlier ideas of neural history compression (Schmidhuber, 1992a). While promising, the PHi framework introduces significant practical challenges: it requires inserting a noisy information bottleneck that can degrade model performance, needs further model training which can be unstable, and is highly sensitive to the precise location of its placement within the model, and the weighting of multiple loss terms.

In this work, we propose a simplified and more direct measure, Multiple Token Divergence (MTD), which quantifies information gain in the model's output distribution. The core insight is simple: if a shallow computational shortcut (e.g., a single Transformer block) can approximate the full model's

prediction, then the model is not performing particularly complex computation. If, however, there is a significant divergence between these two predictions, we can conclude that the model is leveraging its deeper computational capacity. MTD is straightforward to implement and can even be computed directly using the Multiple Token Prediction (MTP) modules that some modern pre-trained models already possess, requiring no additional fine-tuning. In addition to this measure, we present Divergence Steering, a novel decoding method that uses the MTD signal to control the computational character of the generated output. It biases the decoding distribution towards or away from computationally dense tokens—i.e., tokens that are likely under the full prediction but unlikely under shallow approximation by the MTP module. We empirically demonstrate that MTD is more effective than prior methods at distinguishing complex difficult tasks from simple ones. We also investigate the properties of MTD and Divergence Steering in reasoning and creative generation tasks. Our implementation is public (github.com/vincentherrmann/multiple-token-divergence).

## 2 BACKGROUND

### 2.1 PREDICTION OF HIDDEN STATES (PHI)

The Prediction of Hidden states (PHi) method, introduced by Herrmann et al. (2025), creates an information bottleneck (Tishby and Zaslavsky, 2015) within a sequence model in order to measure the complexity of its in-context computation. A PHi layer is inserted between a model's "bottom" and "top" layers. The model consists of the following modules: The **Bottom Layers** ($B_\beta$) are the initial Transformer blocks that process the input sequence embeddings. The **PHi Layer** contains three key components: (1) An **encoder** ($q_\psi$) that, at time step $t$, maps the hidden state $g_t$ from the bottom layers to a posterior distribution over a latent variable $z_t$. This distribution is typically a diagonal Gaussian, similar to variational auto-encoders (Kingma and Welling, 2014; Rezende et al., 2014). (2) A **de-**

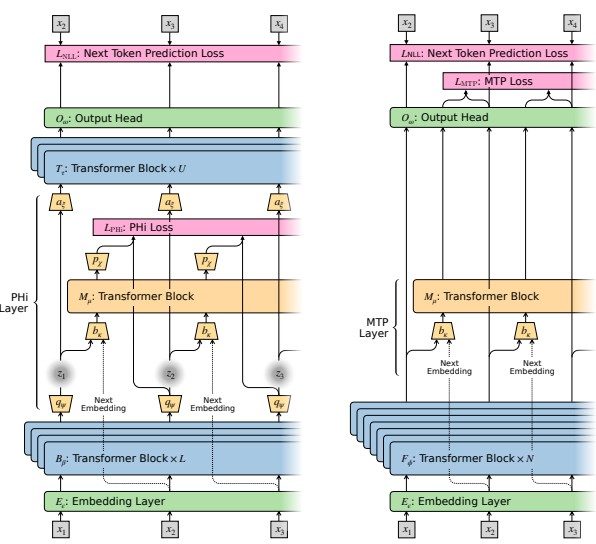

Figure 1: Comparison between the architecture of a PHi model (left) and of a MTP model (right).

**coder** ($a_\xi$) that reconstructs the hidden state, creating $g'_t$ from a sample of the latent variable $z_t$. (3) An **autoregressive prior** that predicts the distribution of the current latent $z_t$ using only the history of previous latents, $z_{<t}$. This can be implemented with a single Transformer block ($M_\mu$) and two additional linear transforms: $b_\kappa$, which maps the inputs to $M_\mu$ to the right dimensionality, and $p_\chi$, which maps the output of the Transformer to a prior distribution. The **Top Layers** ($T_\tau$) are the remaining Transformer blocks that process the sequence of reconstructed hidden states $g'$. Finally, we have the standard token **Embedding** ($E_\epsilon$) and the **Output Layer** ($O_\omega$). Here and in the remainder of the paper, Greek subscript letters indicate learnable neural network parameters.

The forward pass processing tokens $x_1, x_2, \ldots$ is described by these equations:

$$e_t = E_\epsilon(x_t) \qquad \text{Token embedding}$$
$$g_t = B_\beta(e_1, \ldots, e_t) \qquad \text{Output from bottom layers}$$
$$z_t \sim q_\psi(\,\cdot\,|g_t) \qquad \text{Latent sample from posterior}$$
$$g'_t = a_\xi(z_t) \qquad \text{Reconstruction of hidden state}$$
$$h_t = T_\tau(g'_1, \ldots, g'_t) \qquad \text{Output from top layers}$$
$$\pi(\,\cdot\,|x_{<t}) = O_\omega(h_{t-1}) \qquad \text{Next token prediction from output head}$$
$$L_{\text{NLL}}(t) = -\log \pi(x_t|x_{<t}) \qquad \text{Negative Log Likelihood (NLL) loss}$$

The PHi loss ($L_{\text{PHi}}$) is the KL divergence between the posterior $q_\psi$, which has access to the current input $x_t$ via $g_t$, and the prior $p_\chi$, which only has access to past latents $z_{<t}$. We assume an initial latent $z_0$ is given.

$$c_t = b_\kappa(z_{t-1}) \qquad\qquad \text{Linear projection of last latent} \qquad\qquad (1)$$
$$d_t = M_\mu(c_1, \ldots, c_t) \qquad\qquad \text{Output from PHi Transformer block}$$
$$L_{\text{PHi}}(t) = D_{\text{KL}}\Big(q_\psi(\,\cdot\,|g_t)\,||\,p_\chi(\,\cdot\,|d_t)\Big) \qquad\qquad \text{PHi Loss} \qquad\qquad (2)$$
$$= D_{\text{KL}}\Big(q_\psi(\,\cdot\,|x_1, \ldots, x_t)\,||\,p_\chi(\,\cdot\,|z_0, z_1, \ldots, z_{t-1})\Big)$$

The model is trained to jointly minimize both $L_{\text{NLL}}$ and $L_{\text{PHi}}$. The PHi loss quantifies the information gain at each timestep—the amount of new useful information present in the current input token $x_t$ that was not predictable from the history of latent states. It has been shown (Herrmann et al., 2025) that this value correlates well with the complexity and "interestingness" of tasks.

## 2.2 Multiple Token Prediction (MTP)

Multiple Token Prediction (MTP) is a technique used to improve model performance and enable faster inference via methods like speculative decoding (Cai et al., 2024; Gloeckle et al., 2024; Liu et al., 2024; Xiaomi et al., 2025). In this setup, a computationally cheap auxiliary module is trained to directly predict the main model's future output distribution.

First, consider a standard autoregressive model's forward pass:

$$e_t = E_\epsilon(x_t) \qquad\qquad \text{Token embedding}$$
$$h_t = F_\phi(e_1, \ldots, e_t) \qquad\qquad \text{Output of all main Transformer blocks}$$
$$\pi(\,\cdot\,|x_{<t}) = O_\omega(h_{t-1}) \qquad\qquad \text{Next token prediction from output head}$$
$$L_{\text{NLL}}(t) = -\log \pi(x_t|x_{<t}) \qquad\qquad \text{NLL loss}$$

The goal of MTP is to approximate the main model's prediction for the token one step further ahead, $x_{t+1}$. Note that often in MTP, there are additional modules that approximate predictions for tokens even further ahead, i.e., $x_{t+n}$ for $n > 1$. We will not use them in this work.

A separate, smaller MTP module (e.g., a single Transformer block $M_\mu$) generates its own prediction without access to the full model's current hidden state $h_t$ and is usually trained with a negative log-likelihood loss, which we call $L_{\text{MTP}}$. Optionally, the MTP module can be given access to the current token embedding $e_t$, as indicated by the square brackets.

$$c_t = b_\kappa(h_{t-1}[, e_t]) \qquad\qquad \text{Input to MTP module (projection of } h_{t-1} \text{and possibly } e_t) \quad (3)$$
$$d_t = M_\mu(c_1, \ldots, c_t) \qquad\qquad \text{Output from MTP Transformer block}$$
$$\pi_{\text{MTP}}(\,\cdot\,|x_{<t}) = O_\omega(d_{t-1}) \qquad\qquad \text{MTP's prediction for token } x_{t+1}$$
$$L_{\text{MTP}}(t) = -\log \pi_{\text{MTP}}(x_t|x_{<t}) \qquad \text{MTP Loss}$$

In order to predict the next token $x_{t+1}$, the MTP module has access to the model's history via $h_{t-1}$ (and optionally the current embedding $e_t$), but it crucially lacks the result of the main model's full computation at step $t$ (i.e., $h_t$). In recent works using MTP for Large Language Models (Xiaomi et al., 2025; Liu et al., 2024), the MTP module shares the embedding and output heads with the original model, implicitly enforcing alignment of the latent space.

## 3 Multiple Token Divergence

Observe the similarity between the PHi framework's autoregressive prior $p_\chi$ and posterior $q_\psi$ on the one hand, and the MTP prediction $\pi_{\text{MTP}}$ and the full model prediction $\pi$ on the other (Figure 1): in both cases, a computationally and informationally constrained module approximates the prediction from a full model. The key difference is that for PHi, this approximation occurs in a continuous latent space, whereas MTP operates directly on the discrete token distribution.

Based on this analogy, we propose the *Multiple Token Divergence* (MTD) as an alternative to the PHi loss. It is defined as the KL divergence between the full model's next-token prediction $\pi$ and the MTP module's prediction $\pi_{\text{MTP}}$:

$$
\begin{aligned}
L_{\text{MTD}}(t) &= D_{\text{KL}}\Big( \pi( \, \cdot \, | x_{\leq t}) \, || \, \pi_{\text{MTP}}( \, \cdot \, | x_{<t}) \Big) \qquad\qquad (4)\\
&= D_{\text{KL}}\Big( \pi\big( \, \cdot \, | F_\phi(e_1, \ldots, e_t) \big) \, || \, \pi_{\text{MTP}}\big( \, \cdot \, | F_\phi(e_1, \ldots, e_{t-1})[, e_t] \big) \Big).
\end{aligned}
$$

The MTD loss, $L_{\text{MTD}}$, can either be optimized directly in conjunction with the standard next-token loss $L_{\text{NLL}}$, or it can be calculated post-hoc using an MTP module trained with $L_{\text{MTP}}$ loss (see Section 4.2). For now, we ignore the optional access of the MTP module to the latest embedding $e_t$; this will be addressed below.

**On the Difference between PHi and MTD**    While PHi introduced a powerful conceptual framework for analyzing a model's internal processing, its implementation can be complex. The stochasticity introduced by the variational information bottleneck can also interfere with the main sequence prediction task. In contrast, MTD is significantly simpler to implement as it functions as a non-invasive auxiliary task; providing a more direct and less disruptive method to obtain similar insights into the model's per-token computational effort.

One interpretation of the PHi loss is that it measures, at every step, changes of the "latent program" that the model synthesizes in-context to perform next-token prediction. The MTD loss, however, measures changes directly at the level of the output predictions. This distinction can lead to significant differences. For instance, a small change in the latent program could result in a large shift in the output predictions. As an illustrative example, consider a model trained on two distinct types of sequences: one type consists of uniformly random tokens, while the other is a specific single, repeated token. To distinguish between these two cases, the latent program only needs to gain one bit of information. The PHi loss would therefore be low. However, the resulting change in the output distribution is large—shifting from a uniform distribution to a one-hot distribution. In this scenario, the MTD can be as high as $D_{KL}(\text{one-hot}||\text{uniform}) = \log_2(\text{vocabulary size})$ bits. In such cases, we expect $L_{\text{MTD}}$ to be significantly higher than $L_{\text{PHi}}$, an effect we observe in our experiments in Section 4.1. Conversely, one can imagine cases where the latent program changes significantly while the output predictions remain stable. However, the PHi training objective penalizes encoding such changes, as they do not sufficiently improve downstream predictions and thus represent an inefficient use of the information bottleneck. Whether the change in the latent program is larger than the one in the output distributions or not depends on the exact weighting of the PHi loss during training.

**Access to the Latest Token Embedding**    An interesting nuance for both PHi loss and MTD is that the measured information gain at step $t$ can originate from two distinct sources: (1) novel information contained within the current token $x_t$ itself, and (2) complex computation performed by the model's main layers ($B_\beta$ for PHi, $F_\phi$ for MTP), which cannot be easily approximated by the simpler prior or MTP module ($M_\mu$). To disentangle these two sources, we can provide the prior/MTP module with direct access to the latest token embedding $e_t$, which is a common practice in MTP models (see Equation 3). This effectively isolates the second source of information gain. For the PHi framework, this modification involves updating Equation 1 to concatenate the previous latent state with the current embedding:

$$
c_t = b_\kappa(z_{t-1}, e_t).
$$

With this change, the PHi prior has access to the same input token as the bottom layers, $B_\beta$. Consequently, the modified PHi loss, which we denote as $\hat{L}_{\text{PHi}}$, isolates the information gain attributable solely to the computation performed by $B_\beta$:

$$
\hat{L}_{\text{PHi}}(t) = D_{\text{KL}}\Big( q_\psi( \, \cdot \, | x_1, \ldots, x_t) \, || \, p_\chi( \, \cdot \, | z_0, \ldots, z_{t-1}, e_t) \Big).
$$

A PHi layer modified in this way acts as an information bottleneck that specifically measures computational effort. Information that the prior can easily extract from the input embedding $e_t$ is allowed to pass freely, while information that is computationally non-trivial for $B_\beta$ to extract is quantified by $\hat{L}_{\text{PHi}}$. The same logic applies to the MTD module when it is given access to the latest embedding.

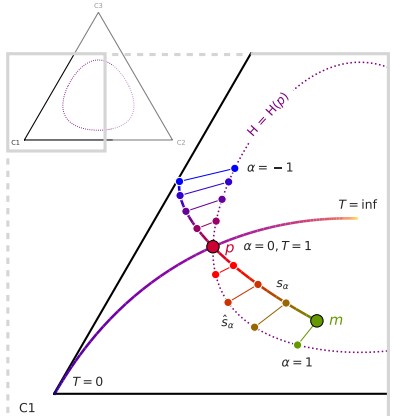
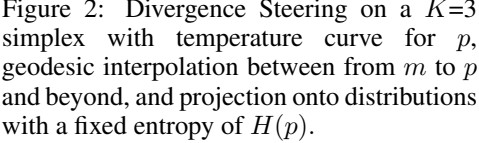
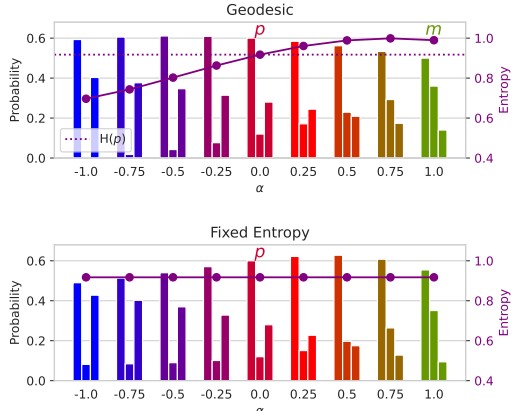

Figure 2: Divergence Steering on a $K$=3 simplex with temperature curve for $p$, geodesic interpolation between from $m$ to $p$ and beyond, and projection onto distributions with a fixed entropy of $H(p)$.

Figure 3: Distributions corresponding to Figure 2. Geodesic interpolation $s_\alpha$, and the entropy of the resulting distribution (top). The same distributions projected onto the surface with fixed entropy, $\hat{s}_\alpha$ (bottom).

Arguably, PHi and MTD loss with access to the latest embedding provide a better measure of dense in-context computation. This access allows the prior/MTP module to account for trivial shifts in the predictions—like the one described in Section 3—thereby reducing the effective difference between the two metrics. In essence, providing access to the latest embedding allows us to quantify the information gain per step that is due to significant computational effort, whether measured in the latent space (PHi) or in the output distribution (MTD).

### 3.1 DECODING WITH DIVERGENCE STEERING

So far, we have presented MTD as a post-hoc analysis tool. However, its formulation, based on the divergence between two output distributions, provides a mechanism to influence the model's behavior during generation. This allows us to steer the decoding process towards or away from tokens that the shallow MTP module can easily predict. This gives rise to a novel decoding method.

The core idea is to construct a new sampling distribution, $s_\alpha$, by interpolating between the full model's prediction, $\pi$, and the MTP module's prediction, $\pi_{\mathrm{MTP}}$. This is controlled by a single parameter, $\alpha$: For $\alpha = 0$, we recover the original distribution from the full model: $s_0 = \pi$. For $\alpha = 1$, we use the distribution from the shallow MTP module: $s_1 = \pi_{\mathrm{MTP}}$. For $\alpha < 0$, we extrapolate away from the MTP module's prediction. This amplifies the probability of tokens that are considered likely by the full model but unlikely by the shallow shortcut, effectively creating an "anti-speculative" distribution biased towards computationally intensive tokens.

To perform this interpolation in a principled way, we travel along the geodesic path between the two distributions under the Fisher-Rao metric. This is achieved by mapping the distributions onto the positive orthant of a hypersphere and performing spherical linear interpolation (Miyamoto et al., 2024). Let $p = \pi$ and $m = \pi_{\mathrm{MTP}}$ be two categorical distributions over a vocabulary of size $K$. Their representations on the hypersphere are the square roots of their probabilities:

$$\mathbf{p}_g = (\sqrt{p_1}, \sqrt{p_2}, \dots, \sqrt{p_K})$$
$$\mathbf{m}_g = (\sqrt{m_1}, \sqrt{m_2}, \dots, \sqrt{m_K})$$

The angle between these two vectors is $\Theta = \arccos\left(\sum_k \sqrt{p_k m_k}\right)$. The geodesic path $\mathbf{s}_g(\alpha)$ between $\mathbf{p}_g$ and $\mathbf{m}_g$ is then given by:

$$\mathbf{s}_g(\alpha) = \frac{\sin((1-\alpha)\Theta)}{\sin(\Theta)}\mathbf{p}_g + \frac{\sin(\alpha\Theta)}{\sin(\Theta)}\mathbf{m}_g$$

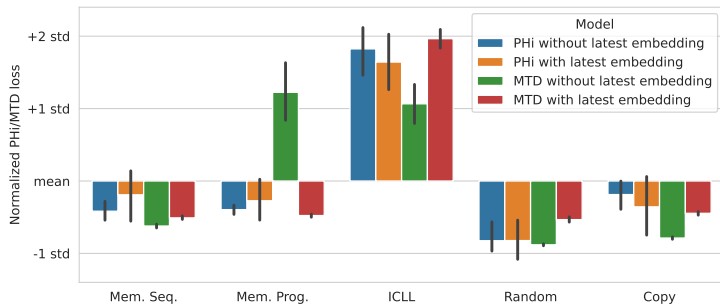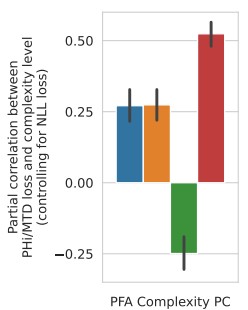

Figure 4: Normalized PHi or MTD loss of the four different model types on each of the five tasks. Only in-context language learning (ICLL) requires sophisticated in-context computation. This is reflected by the scores, with the exception of the MTD model without access to the latest embedding, which assigns high MTD also to the memorized programs task (see the discussion in Sections 3 and 4.1). Bootstrapped mean with $95\%$ confidence intervals across 8 runs.

Figure 5: Partial correlation of PHi or MTD loss with the complexity of the modelled PFA, controlling for NLL. Also here, MTD without latest embedding access is the outlier.

To map this path back to a valid probability distribution $s_\alpha$, we square each component of the vector $\mathbf{s}_g(\alpha)$, i.e., $s_k(\alpha) = (\mathbf{s}_{g,k}(\alpha))^2$.

This method introduces a new control knob, $\alpha$, which is complementary to the standard temperature parameter, $T$. While $T$ adjusts the entropy of the output distribution, $\alpha$ adjusts its "computational character." Because $\pi_{\mathrm{MTP}}$ often has higher entropy than $\pi$, changing $\alpha$ can also affect entropy. To isolate these effects, we can optionally project the interpolated distribution $s_\alpha$ to a new distribution $\hat{s}_\alpha$ such that its entropy matches that of the original distribution, i.e., $H(\hat{s}_\alpha) = H(\pi)$ for all $\alpha$. This provides two orthogonal levers for shaping the decoding process: $T$ for entropy and $\alpha$ for computational density. Figures 2 and 3 visualize the method, additional details can be found in Appendix A. As we will show, the optimal choice of $\alpha$ is task-dependent (which is also the case for $T$): some tasks benefit from the robust, simpler predictions favored by positive $\alpha$, while others may require the novel, less obvious paths uncovered by negative $\alpha$.

## 4 EXPERIMENTS

### 4.1 MTD AND PHI LOSS OF SEQUENCE MODELS TRAINED FROM SCRATCH

The considerations from Section 3 leave us with four different model configurations to compare: PHi and MTD models, each with and without access to the latest token embedding. The PHi model without this access corresponds to the original method proposed in prior work (Herrmann et al., 2025). To compare these different setups, we train Transformer models from scratch on several tasks and evaluate them in settings similar to those in (Herrmann et al., 2025). For details on the exact training setups, please see Appendix B.1.

**Evaluation on Different Tasks** The four model types are trained on five different tasks: (1) reciting memorized sequences, (2) modeling sequences from a small set of known formal languages (memorized programs), (3) in-context language learning (ICLL), where the formal language is unknown (Akyürek et al., 2024), (4) modeling random token sequences, and (5) a copying task that involves modeling random tokens where subsequences appear twice. Of these, only ICLL—which requires inferring the structure of an unknown probabilistic finite automaton (PFA) in-context—involves meaningful computation, in the sense that a non-trivial latent program must be synthesized by the model. Figure 4 shows a comparison of the normalized PHi and MTD losses for each task. The MTD with latest embedding access shows the clearest distinction between the one complex task and the four "boring" ones; note the high value for ICLL and the consistently low values for all other tasks. For the MTD model *without* latest embedding access, we see the effect alluded to in Section 3: the loss is high for both ICLL and the memorized programs. For the memorized programs task, the actual in-context program required is minimal (only $\sim \log_2(10)$ bits

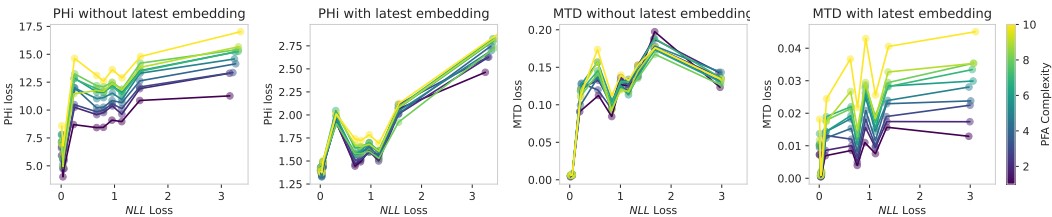

Figure 6: Token-wise PHi loss and MTD against binned NLL, for the different modeled PFA complexities. PHi loss without and MTD with latest embedding access both show a clear correlation with complexity level, across NLL bins. See Figure 9 (Appendix) for a bin-wise normalized version.

to identify any one out of the ten memorized automata). However, the lack of information from the latest token causes a significant shift in the model's output distribution, resulting in a high MTD. Finally, giving the PHi layer access to the latest embedding does not appear to improve its ability to distinguish "boring" from interesting tasks.

**Task Complexity** Focusing on the ICLL task, we investigate the relationship between the models' PHi or MTD losses and the complexity of the underlying language, as measured by the description length of the PFA. Figure 5 displays the partial correlation between the mean PHi or MTD loss across a sequence and the language's complexity. We control for the mean NLL, as it is positively correlated with language complexity (r=0.367, 95% CI [0.315, 0.424]). Here again, we find that MTD with latest embedding access shows the strongest positive correlation (r=0.524 [0.480, 0.565]). In contrast, MTD without access to the latest embedding is negatively correlated with language complexity when controlling for NLL, confirming that it is not a reliable measure for this purpose. Figure 6 shows the token-wise PHi or MTD loss against binned NLL loss, broken down by PFA complexity (from 1, simple, to 10, complex). This analysis reveals that only the original PHi loss (without embedding access) and the MTD loss with embedding access show a clear, positive token-wise relationship with language complexity after controlling for NLL.

## 4.2 Pre-Trained Language Models

To validate our hypotheses on existing large-scale models, we leverage the pre-trained, open-source MiMo-7B model (Xiaomi et al., 2025). We choose it as a high-quality 7B parameter model, whose base pre-training incorporates an MTP objective, providing the built-in auxiliary prediction heads necessary for calculating the MTD without any post-hoc modification. Additionally, we use the Mistral 7B model (Jiang et al., 2023), which provides no MTP module out-of-the-box. Instead we train MTP heads of different sizes by keeping the base model frozen and using standard teacher-student distillation (Schmidhuber, 1992b; Hinton et al., 2015) with a small fraction of the original data and compute (see Appendix B.2). For clarity, in the main part of the paper, we report the results only for the MiMo model. The qualitatively very similar results for the Mistral models can be found in Appendix C.1.

**Reasoning Difficulty** We employ the MATH dataset (Hendrycks et al., 2021b), which provides mathematics problems labeled from Level 1 (easy) to Level 5 (hard), along with detailed reasoning solutions. We first compute the mean MTD for the provided step-by-step solution for each problem in the dataset and find that it clearly correlates with the difficulty level (r=0.179, 95% CI [0.152, 0.203]). Interestingly, the NLL loss *negatively* correlates with problem difficulty (r=-0.249 [-0.274, -0.224]). This suggests that from the model's perspective, reasoning chains for difficult problems are no less plausible or predictable. However, the higher MTD indicates that the model makes increased use of its full capacity to process and generate them. We also have the model generate ten different chains-of-thought (CoTs) for each problem and repeat the analysis on these self-generated solutions. There again, we observe very similar results: the partial correlation between MTD and difficulty level, controlling for NLL, is r=0.199 [0.189, 0.208], while the correlation between NLL and difficulty is r=-0.158 [-0.168, -0.149]. These effects hold consistently across most problem categories, as shown in Figures 10 and 11 (Appendix). Since the provided rationales, as well as

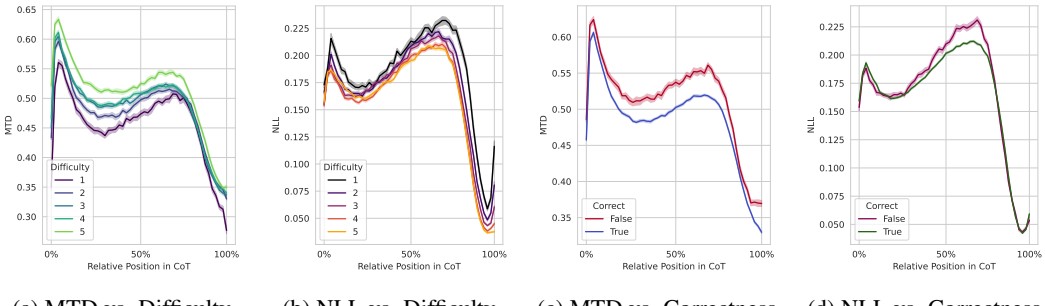

(a) MTD vs. Difficulty    (b) NLL vs. Difficulty    (c) MTD vs. Correctness    (d) NLL vs. Correctness

Figure 7: Token-wise losses against relative positions in self-generated CoT for the MATH test dataset. MTD shows a clear correlation with difficulty across the full CoT (a), the relationship between NLL and difficulty is less clear (b). Similarly, correct CoTs show higher MTD over all relative positions (c), which is not the case for NLL (d).

the generated CoTs, are longer for more difficult problems, the cumulative NLL also correlates positively with difficulty level (see Figures 12 and 13 in the Appendix).

**Token-wise Development Across the Response**  We track the token-wise values for MTD and NLL across each generated CoT. As seen in Figure 7a, the positive correlation between MTD and problem difficulty holds consistently from the first tokens of the response to the last. Likewise, the negative correlation for NLL persists throughout the generation, even though not as pronounced (Figure 7b). The difference between MTD and NLL in their correlation with the problem difficulty is notable because, at a global level, MTD and NLL are positively correlated with each other (r=0.255 [0.246, 0.265]). This highlights that MTD captures a distinct signal related to computational effort that is not present in the standard NLL loss.

**Reasoning Accuracy**  For the self-generated CoTs, we also investigate the relationship between MTD values and the correctness of the final answer. Figure 7c plots the token-wise MTD, stratified by whether the rationale was correct or incorrect. We observe that correct responses are consistently associated with lower MTD. The relationship between NLL and correctness is less consistent (see Figure 7d). Following the methodology from prior work (Herrmann et al., 2025), we randomly assemble pairs of one correct and one incorrect CoT for each math problem. The probability of choosing the correct CoT when picking the one with the lower mean MTD is 67.1% (95% CI: [65.4%, 68.7%]). When selecting the one with the lower NLL, the probability is 73.3% [71.9%, 74.8%]. For the cases where NLL and MTD are agree, we get 80.4% [78.5%, 81.9%] accuracy. We repeat these experiments on the GSM-8k dataset (Cobbe et al., 2021), where we find that selecting CoTs with lower MTD yields 66.0% [62.9%, 69.2%] correct answers, while lower NLL yields 72.2% [69.1%, 75.0%] and combined yields 75.5% [71.7%, 79.2%]. For token-wise curves, please see Appendix C.1. These findings stand in contrast to the results for PHi loss, where, for a Llama 3B model, correct answers are associated with a *high* PHi loss (Herrmann et al., 2025). While further investigation is needed, we hypothesize that different models may have different tendencies to either overly simplify or overly complicate their reasoning process (Sui et al., 2025). This tendency could determine whether computationally intensive answers—as opposed to more straightforward ones—are more or less likely to be correct for a given model architecture or training regime.

## 4.3  DIVERGENCE STEERING AND CREATIVE TASKS

Having established MTD as an indicator of complex in-context computation, we now investigate whether we can use it to influence model generation. Specifically, can biasing generation towards tokens with high MTD lead to more complex or creative outputs? The Divergence Steering method allows us to test this hypothesis.

**Algorithmic Toy Tasks**  We adopt the framework proposed by Nagarajan et al. (2025), training tTransformer models on four distinct tasks: sibling discovery, triangle discovery, circle construc-

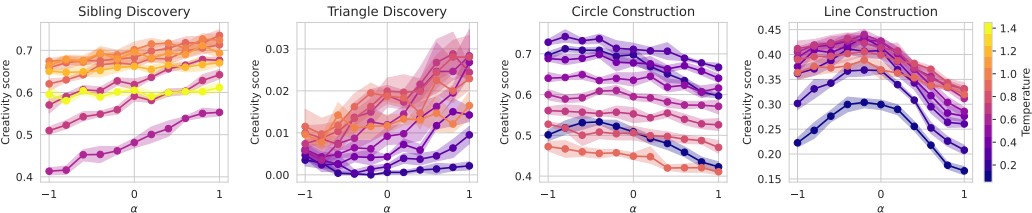

Figure 8: For the discovery tasks, positive $\alpha$ leads to higher creativity, whereas for the construction tasks, negative $\alpha$ leads to higher creativity. Results for geodesic distributions $s_\alpha$.

tion, and line construction (for details, please see Appendix B.3). The objective for each task is to generate sequences that are simultaneously *valid*, *novel* (i.e., not memorized from the training set), and *unique* within a fixed number of attempts. Success is measured by a creativity score, where 1 indicates perfect performance across all three criteria. All models used in this experiment are configured with MTP modules that have access to the latest token embedding. Figure 8 shows the creativity scores across a range of values for temperature and our steering parameter, $\alpha$. The results reveal a task-dependent effect. For the "discovery" tasks, positive values of $\alpha$—which bias generation toward the simpler predictions of $\pi_{\mathrm{MTP}}$—yield higher creativity scores. For the "construction" tasks, negative values of $\alpha$—which create an "anti-speculative" distribution biased away from $\pi_{\mathrm{MTP}}$—lead to better performance. A more detailed analysis (Appendix C.2) suggests that positive $\alpha$ helps the model avoid memorized solutions (improving novelty), whereas negative $\alpha$ can encourage the generation of more structurally sound outputs (improving validity). The optimal strategy, therefore, depends on the specific demands of the task. Crucially, temperature and $\alpha$ function as largely independent controls over the decoding process: for all four tasks, the best-performing combination of temperature and $\alpha$ achieves significantly higher creativity scores than optimizing for temperature alone. The qualitative behavior is similar for both geodesic and fixed-entropy distributions (see Figure 16 in the Appendix).

**Creative Writing Benchmark** We also investigate how Divergence Steering can affect the creative writing performance of the MiMo model. For the creative writing benchmark (Paech, 2025), the model generates several short stories following various prompts. An LLM judge scores the generated stories according to many different criteria, and also assigns a main score, the "Overall Impression". From Table 1, we can see that a slightly negative steering parameter ($\alpha = -0.1$) leads to the best results overall. Most criteria, including Overall Impression, reach the best scores when biasing tokens away from the MTP distribution via Divergence Steering. As we would expect, we get the least "Unsurprising and Uncreative" stories for negative $\alpha$. For positive $\alpha$, however, stories show fewer characteristics connected with overcomplication, such as "Purple Prose" and "Overwrought". The experimental details and additional results can be found in Appendix B.3 and C.3. These results show that shaping the computational character of the decoding distribution can have positive effects, especially for creative tasks.

## 5 DISCUSSION & FUTURE WORK

In our experiments, MTD outperforms PHi loss in differentiating "boring" from "interesting" tasks and simple from complex ones. It successfully isolates the per-token information gain attributable to non-trivial, or "irreducible" (Wolfram, 2002) computation by the model. It should be noted that the utility of the MTD signal can be contingent on the relative capacities of the main model and the MTP module ($M_\mu$): if the MTP module is too powerful, MTD approaches zero, and if it is too weak, MTD offers little beyond the standard NLL loss. However, experiments with several different MTP sizes show robustness of the effects reported in this work (Appendix C.1). Furthermore, MTD may conflate genuine computational effort with simple patterns that exceed the capacity of the lightweight shortcut.

Our findings also surface several intriguing questions. The positive correlation of MTD with problem complexity, in direct contrast to the negative correlation of NLL, warrants further investigation to determine if this is a general pattern across models and scales. Similarly, our result that lower

Table 1: Creative writing criteria which reach the best score for a given $\alpha$ value. For 10 different criteria, including Overall Impression, $\alpha = -0.1$ gives the best results. In contrast, $\alpha = 0$ works best only for 6 criteria. Criteria followed by a down arrow ($\downarrow$) are negative, meaning that a low score is best. Results for geodesic distributions $s_\alpha$.

| $\alpha = -0.4$ | $\alpha = -0.2$ | $\alpha = -0.1$ | $\alpha = 0.0$ | $\alpha = 0.1$ | $\alpha = 0.2$ | $\alpha = 0.4$ |
|---|---|---|---|---|---|---|
| — | (1) Unsurprising or Uncreative $\downarrow$ | **(1) Overall Impression**, | (1) Overall Reader Engagement, | (1) Unearned Transformations $\downarrow$, | — | (1) Overwrought $\downarrow$ |
| | | (2) Emotionally Complex, | (2) Consistent Voice/Tone of Writing, | (2) Incongruent Ending Positivity $\downarrow$ | | (2) Purple Prose $\downarrow$ |
| | | (3) Emotionally Engaging, | (3) Coherent, | | | |
| | | (4) Adherence to Instructions, | (4) Sentences Flow Naturally, | | | |
| | | (5) Believable Character Actions, | (5) Amateurish $\downarrow$, | | | |
| | | (6) Nuanced Characters, | (6) Meandering $\downarrow$ | | | |
| | | (7) Imagery and Descriptive Quality, | | | | |
| | | (8) Elegant Prose, | | | | |
| | | (9) Well-earned Lightness or Darkness, | | | | |
| | | (10) Weak Dialogue $\downarrow$ | | | | |

MTD is associated with correct reasoning contrasts with prior findings for PHi loss, suggesting the relationship between computational effort and correctness is complex and model-dependent. While Divergence Steering enhanced performance on creative tasks, in preliminary experiments we found no clear improvement in the reasoning of large pre-trained models, perhaps because significant changes to the decoding strategy interfere with behaviors learned during post-training.

Our findings suggest that MTD and Divergence Steering have the potential for many applications in training and inference. Examples could be **Dynamic Compute Allocation:** MTD could be monitored in real-time during generation. A prolonged period of low MTD might trigger early stopping for a simple task, while a sudden spike in MTD could activate more powerful components (e.g., additional Mixture-of-Experts layers) for a difficult step. **Solution Convergence:** The transition from a high-MTD processing phase to a low-MTD conclusion could act as a signal that the model has "settled" on a solution, potentially allowing for more efficient decoding. **Intrinsic Motivation:** In agent-based settings, MTD could serve as an intrinsic reward. This would encourage an agent to pursue policies that lead to computationally interesting states (high information gain), fostering the development of more sophisticated behaviors. **Open-Endedness:** MTD and Divergence Steering allows the filtering or direct generation of "interesting" data. This may help to prevent model collapse when training on self-generated data and enable more creative, open-ended learning.

## 6 CONCLUSION

In this work, we introduce Multiple Token Divergence (MTD), a practical and direct measure for quantifying the computational effort of language models. By measuring information gain in the output distribution, MTD serves as a more robust and stable metric than prior methods that rely on latent state compression. We show that giving the auxiliary prediction module access to the latest token embedding allows MTD to specifically isolate the information gain attributable to non-trivial computation. Our findings demonstrate that MTD successfully distinguishes complex in-context reasoning from simpler tasks and reveals a nuanced relationship between computational effort and predictive loss. As a non-invasive and easily implemented metric, MTD provides a valuable new tool for analysis and evaluation. Furthermore, we introduce Divergence Steering, a novel decoding method that uses the MTD signal to actively steer the generation process towards either more or less computationally dense sequences. Shaping this "computational character" is complementary to the standard entropy adjustment using decoding temperature and can help with creative tasks.

## REPRODUCIBILITY STATEMENT

All experiments can be reproduced using a single consumer-grade GPU (e.g., NVIDIA GeForce RTX 4090). Appendix B provides the necessary details for the exact replication of our experiments. The weights of the pre-trained models used in this work, MiMo 7B (Xiaomi et al., 2025) and Mistral 7B (Jiang et al., 2023), are publicly available. Similarly, we employ open-source training and test datasets: FineWeb (Penedo et al., 2024), MATH (Hendrycks et al., 2021a), and GSM-8k (Cobbe et al., 2021). The PFA-related tasks (Section 4.1) follow the implementation in Herrmann et al. (2025), and the creative toy tasks (Section 4.3) are adopted from Nagarajan et al. (2025), both with no changes except the ones specified. The prompts used for the creative writing benchmark are also publicly available (Paech, 2025).

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

## A    FIXED ENTROPY PROJECTION FOR DIVERGENCE STEERING

To project the distribution $s_\alpha$ onto the hypersurface with entropy $H(p)$, we solve the following optimization problem:

$$\min_{\hat{s}_\alpha} \quad D_{\text{KL}}(\hat{s}_\alpha || s_\alpha)$$
$$\text{subject to} \quad H(\hat{s}_\alpha) = H(p)$$
$$\sum_i \hat{s}_{\alpha,i} = 1$$

By solving the Lagrangian, we see that this is equivalent to finding a temperature-scaled version of $s_\alpha$. This means that $\hat{s}_\alpha$ takes the form:

$$\hat{s}_\alpha = \text{softmax}\left(\frac{\log s_\alpha}{T}\right)$$

for some temperature T, such that the entropy constraint $H(\hat{s}_\alpha) = H(p)$ is met. Since entropy is a smooth monotonic function of the temperature, we can use a fast root-finding algorithm like binary search to find the correct value for $T$.

In practice, divergence steering, either with geodesic interpolation or this fixed-entropy projection, does not meaningfully slow down the generation process. For large vocabularies, however, it might be sensible use Divergence Steering in combination with top-k sampling and only optimize the remaining smaller distribution.

## B    EXPERIMENT DETAILS

### B.1    SEQUENCE MODELS TRAINED FROM SCRATCH

We train all models using the Adam optimizer, a batch size of 16 and gradient norm clipping of 1.0. The learning rate is 0.0003, with a 500 step linear warm-up from zero and no decay. All losses are weighted equally, for the PHi loss we take the mean of the element-wise KL-Divergence for $z$, not the sum. Every model variation is trained 8 times with different random seeds for the initial weights and the procedurally generated data (which results in different memorized sequences and programs). The training of a model can be done on a single consumer-grade GPU (e.g., NVIDIA RTX 4090).

The base model is based on the Llama 3.2 architecture (Dubey et al., 2024).

- Number of layers: 12
- Model dimensionality: 768
- Number of attention heads: 6
- MLP intermediate size: 2048
- Embedding layer and output head are tied

**PHi models:**

To prevent posterior collapse, we employ an additional contrastive self-critic loss (Menon et al., 2022).

- Training steps: $30,000$
- Placement of the PHi Layer: After the 10th layer
- $z$ dimensionality: 768
- $q_\psi$: Linear transform
- $a_\xi$: Linear transform
- $b_\kappa$: Linear transform
- $M_\mu$: One Transformer block like the ones in the rest of the model
- $p_\psi$: Linear transform

**MTD models:**

- Training steps: $10,000$
- $b_\kappa$: Linear transform
- $M_\mu$: One Transformer block like the ones in the rest of the model

For generation of training and testing data, we follow Herrmann et al. (2025). The only difference is that we do not perturb any tokens during training, and that we use the same models for the task differentiation and task complexity experiments (Section 4.1).

## B.2 Pre-Trained Language Models

For our experiments in Section 4.2, we use the SFT version of the MiMo-7B model (Xiaomi et al., 2025). To calculate the MTD, we use the included MTP head that predicts one token in advance.

For additional experiments, we use the instruction tuned Mistral 7B v0.3 model (Jiang et al., 2023). We train MTP modules of three different sizes: **MTP small** consists of one Transformer block, **MTP medium** of two blocks, and **MTP large** of three. They are initialized using the weights from the last layers of the full model and trained on data from the FineWeb (Penedo et al., 2024), the MATH training set, and the GSM-8k training set (a mixture of 80%, 10% and 10% respectively). We use MTD as the training loss. The MTP modules have access to the latest embedding. We use the AdamW optimizer, a batch size of 4 and train for 30,000 steps, with cosine annealing and a maximum learning rate of 0.0001. As shown in Table 2, this can be done in a few hours on a consumer-grade GPU.

All experimental results include bootstrapped 95% confidence intervals.

## B.3 Divergence Steering and Creativity Tasks

**Algorithmic Toy Tasks**   The MTD models use the architecture and training procedure specified in Section B.1. For each task, a dedicated model is trained for $50,000$ steps. No seed conditioning is used. For task definitions and evaluation procedure, we refer to Nagarajan et al. (2025).

The creativity score is defined as the fraction of all generated items that are valid, unique, and novel. In addition, we define three more scores:

- Validity score: fraction of valid items among all generated items
- Uniqueness score: fraction of unique items among valid generated items
- Novelty score: fraction of novel items among valid unique generated items

These are used in the additional empirical analysis in section C.2.

**Creative Writing**   For the generation of the stories, we use the prompting structure from the creative writing benchmark (Paech, 2025). Divergence steering for temperatures $[0.6, 0.7, 0.8]$ and $\alpha$-values $[-0.4, -0.2, -0.1, 0., 0.1, 0.2, 0.4]$ is evaluated, with geodesic interpolation. For each setting, a total of 96 stories is generated and judged. As a judge model, we use Claude Sonnet 4.5. Again we adhere to the prompting structure and criteria proposed by the benchmark. Every story is evaluated individually, no pairwise ELO-type scoring is used. The detailed results are shown in section C.3.

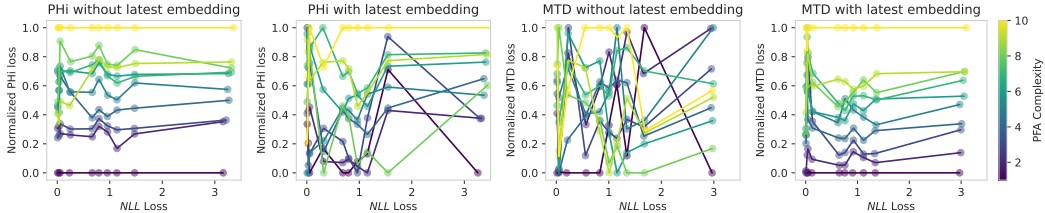

Figure 9: Similar to Figure 6, but normalized for each NLL bin. PHi loss without access to the latest embedding, and MTD loss with access to the latest embedding both show a clear correlation with complexity level, across NLL bins.

Table 2: Comparison between MiMo and Mistral models in terms of tokenizer vocabulary size, loss of the main model (NLL), loss of the MTP module (MTP), multiple token divergence (MTD), and training time. For Mistral models, larger MTP modules lead to reduced MTP Loss and MTD.

| Metric | MiMo | Mistral MTP Small | Mistral MTP Medium | Mistral MTP Large |
|---|---|---|---|---|
| Vocabulary Size | 151936 | 32000 | 32000 | 32000 |
| NLL | 1.345 | 0.943 | 0.943 | 0.943 |
| MTP Loss | 1.425 | 1.298 | 1.217 | 1.178 |
| MTD | 0.469 | 0.553 | 0.476 | 0.429 |
| RTX 4090 hours | — | 9.38 | 10.25 | 11.08 |

## C    ADDITIONAL EXPERIMENTAL RESULTS

Figure 9 shows the normalized PHi losses and MTDs, binned by NLL and normalized, making it clear to see that PHi without and MTD with access to the latest embedding show the clearest token-wise relationship with PFA complexity.

### C.1    DETAILED RESULTS FOR MIMO AND MISTRAL

In this section, we report detailed MTD results for the Mistral 7B models, in comparison with the MiMo model, and for different sizes of the MTP module. The main take-away is that the results are consistent with the findings for MiMo and stable across the MTP module configurations. While qualitatively very similar, they are quantitatively less strong, which suggests that training a model from scratch with an MTP module can have benefits.

Table 2 shows that, as we would expect, with larger MTP module size, the MTP loss and MTD significantly drop. However, as it turns out, this does not affect the relationship between MTD and reasoning difficulty and correctness. We also note that, while due to different tokenization, the losses between MiMo and Mistral models are not directly comparable, MiMo's MTP Loss is much closer to the full NLL, suggesting that its MTP module can predict better than even the large MTP module for Mistral.

From Table 3, we can see that the relationship between task difficulty and MTD is the same for MiMo and Mistral models. For both the provided solutions and for ten self-generated CoTs to the MATH test questions, it consistently shows significant *positive* correlation between MTD and question difficulty (controlled for NLL) and *negative* correlation between NLL and question difficulty (controlled for MTD). The differences in correlation strength for the three MTP module sizes fall within the confidence intervals. For the CoTs generated by Mistral, the partial correlations with difficulty are present and statistically significant, but less clear than for MiMo.

Table 4 shows the relationship between MTD and the correct reasoning: we measure the probability of picking the correct answer out of pair of random pair of generated CoTs, one correct, and the other one incorrect, when choosing based on the specified criterion. We test the MiMo and Mistral models on the test splits of the MATH and GSM-8k datasets. For all models and both datasets, there is a clear connection between low MTD and correctness. This connection goes beyond the relationship between NLL and correctness, which we can see from the fact that the chance of choosing correctly increases significantly if we pick a response with lower NLL *and* lower MTD. Also here, the difference between the Mistral MTP module sizes stay within the confidence intervals.

Table 3: Partial correlation coefficients $r$ for the test questions of the MATH dataset between MTD and difficulty, controlled for NLL, and between NLL and difficulty, controlled for MTD. Results for the provided step-by-step solutions, and for the self-generated CoTs. Across all models, MTD correlates positively with difficulty, and NLL negatively.

| Correlation Type | MiMo | Mistral MTP Small | Mistral MTP Medium | Mistral MTP Large |
|---|---|---|---|---|
| **Provided Solutions** | | | | |
| MTD-Diff (ctrl NLL) | 0.162 [0.137, 0.189] | 0.109 [0.079, 0.138] | 0.094 [0.065, 0.122] | 0.096 [0.066, 0.128] |
| NLL-Diff (ctrl MTD) | -0.239 [-0.266, -0.212] | -0.153 [-0.180, -0.127] | -0.143 [-0.172, -0.115] | -0.144 [-0.173, -0.115] |
| **Generated CoT Solutions** | | | | |
| MTD-Diff (ctrl NLL) | 0.199 [0.189, 0.208] | 0.050 [0.041, 0.058] | 0.044 [0.035, 0.053] | 0.043 [0.034, 0.052] |
| NLL-Diff (ctrl MTD) | -0.205 [-0.215, -0.195] | -0.041 [-0.050, -0.033] | -0.047 [-0.056, -0.038] | -0.047 [-0.056, -0.038] |

Table 4: of the correct answer from pairs of CoTs, for different choosing strategies. Across all models, and for MATH and GSM-8k test questions, lower MTD CoTs are more likely to be correct.

| Criterion | MiMo | Mistral MTP Small | Mistral MTP Medium | Mistral MTP Large |
|---|---|---|---|---|
| **MATH Dataset** | | | | |
| Lower NLL | 73.3% [71.9, 74.8] | 59.1% [57.6, 60.6] | 59.1% [57.6, 60.6] | 59.1% [57.6, 60.6] |
| Lower MTD | 67.1% [65.5, 68.5] | 57.2% [55.6, 58.6] | 57.5% [56.0, 59.0] | 57.8% [56.2, 59.3] |
| Both Lower | 80.4% [78.5, 81.9] | 63.0% [61.1, 64.9] | 62.7% [60.8, 64.6] | 62.7% [60.9, 64.6] |
| **GSM-8k Dataset** | | | | |
| Lower NLL | 72.2% [69.1, 75.2] | 64.9% [63.1, 66.7] | 64.9% [63.1, 66.7] | 64.9% [63.1, 66.7] |
| Lower MTD | 66.0% [62.4, 69.3] | 59.2% [57.5, 61.0] | 61.8% [60.1, 63.5] | 61.3% [59.6, 63.1] |
| Both Lower | 75.5% [71.7, 79.2] | 69.9% [67.6, 72.1] | 71.4% [69.3, 73.5] | 70.9% [68.8, 73.0] |

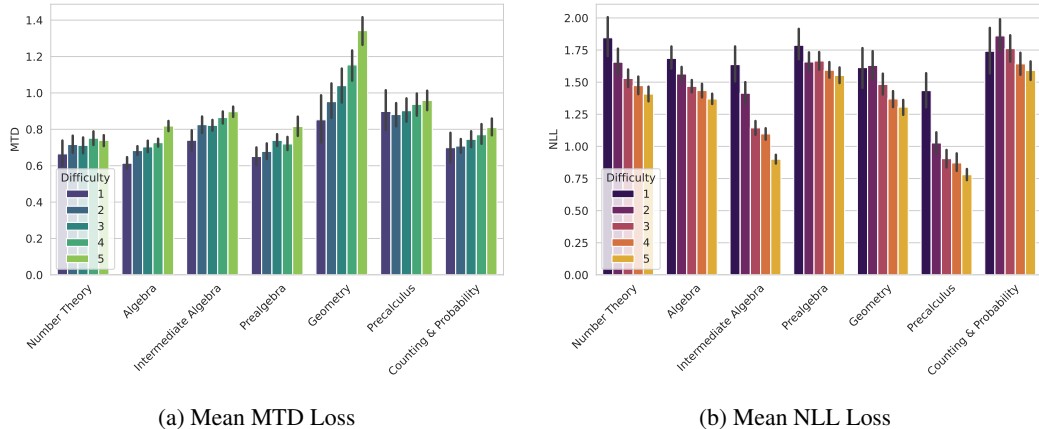

(a) Mean MTD Loss        (b) Mean NLL Loss

Figure 10: Mean losses of the MiMo model across the provided step-by-step solutions to the problems of the MATH test set, grouped by category and difficulty level. MTD clearly grows with difficulty, suggesting that the model is making more use of its computational capacity when processing more challenging problems. NLL loss, on the other hand, goes down with increasing complexity. Figure 11 shows similar results for self-generated chains of thought.

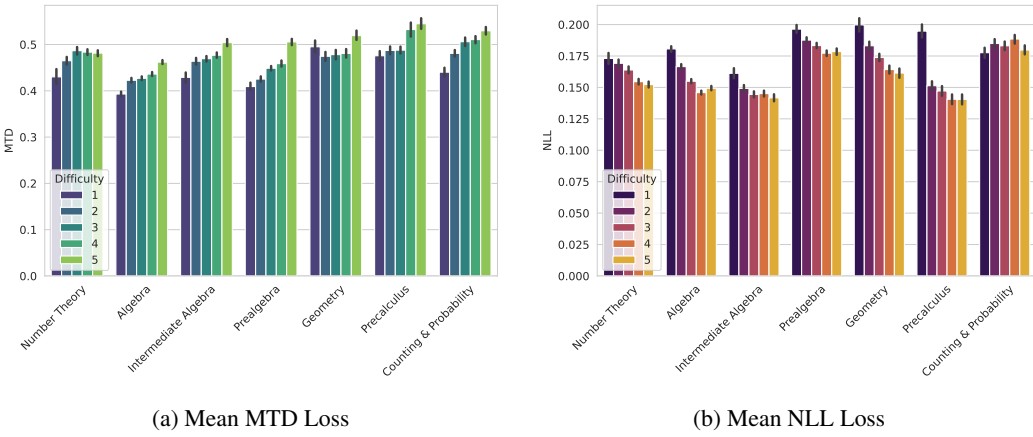

(a) Mean MTD Loss        (b) Mean NLL Loss

Figure 11: Mean losses of the MiMo model across self-generated CoTs for the problems of the MATH test set, grouped by category and difficulty level. Similarly as in Figure 10, we observe that MTD clearly grows with difficulty, as the model is making more use of its computational capacity when generating the solutions to more challenging problems. Also here, the mean NLL goes down with problem difficulty.

**Additional Visualizations** Figure 10 shows MTD and NLL for the provided step-by-step solutions for the MiMo model, broken down by category and difficulty level. Figure 11 shows the same for the self-generated CoTs. The results are qualitatively similar, even though the differences between categories for the CoTs are less pronounced.

Figures 12 and 13 use the cumulative instead of the mean losses. Due to the fact that the provided solutions as well as the generated ones grow in length as the problems become more difficult, cumulative NLL also correlates positively with difficulty level.

Figure 14 shows the development of MTD and NLL across self-generated CoTs for the problems of the GSM-8k test dataset (analogous to Figures 7c and 7d for MATH). Correct CoTs clearly have lower MTD, and lower NLL. Interestingly, for the GSM-8k dataset, the shapes of the NLL curves differ significantly from the shapes of the MTD, missing the prominent initial bump. Currently, we have no explanation for this.

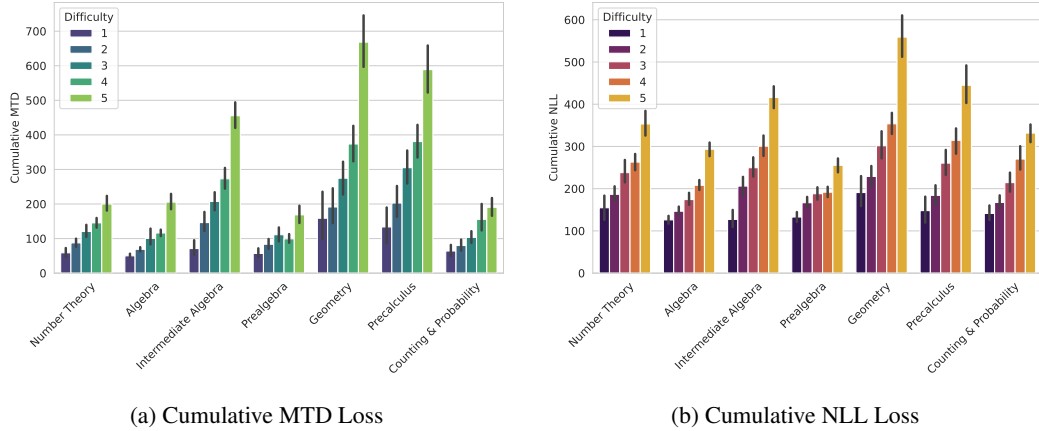

(a) Cumulative MTD Loss

(b) Cumulative NLL Loss

Figure 12: Cumulative losses of the MiMo model across provided solutions from the MATH test set. Since more difficult problems have longer solutions, both cumulative MTD and cumulative NLL correlate with problem difficulty.

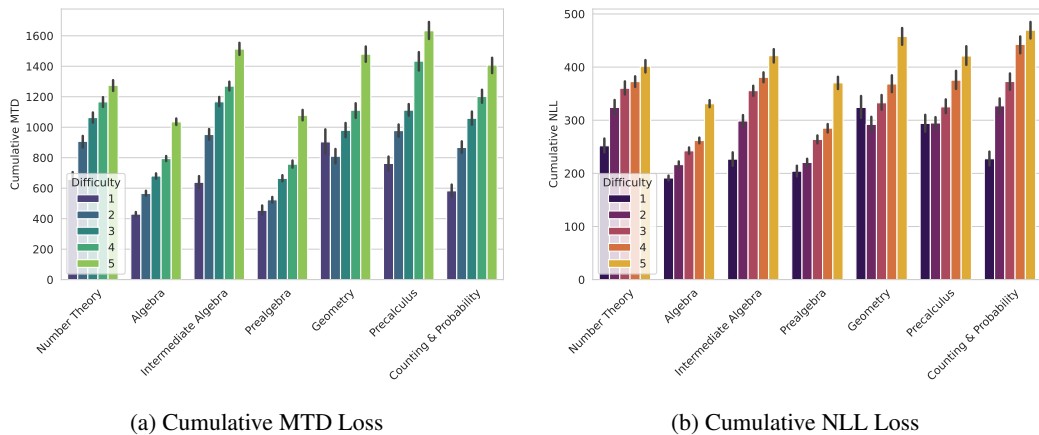

(a) Cumulative MTD Loss

(b) Cumulative NLL Loss

Figure 13: Cumulative losses of the MiMo model across self-generated CoTs for the problems of the MATH test set. We observe a similar effect as in Figure 12.

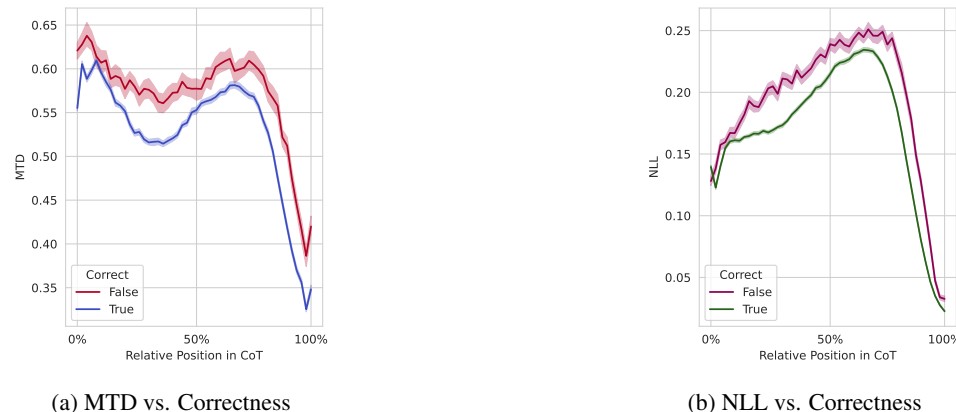

(a) MTD vs. Correctness

(b) NLL vs. Correctness

Figure 14: Token-wise losses against relative positions in self-generated CoTs by MiMo for the GSM-8k test dataset. Lower MTD and lower NLL are both associated with more correct reasoning.

## C.2 DIVERGENCE STEERING AND CREATIVE ALGORITHMIC TASKS

Figure 15 shows the creativity scores for the four tasks, using different values for temperature and $\alpha$. In addition, we break down the results into validity, uniqueness and novelty scores. By the nature of the task, sibling and triangle discovery models are at risk of overfitting to the training data. A positive $\alpha$ value can help avoiding repeating memorized examples, as can be seen from the increased novelty scores. The models for circle and line construction, on the other hand, are less prone to overfitting, due to the combinatorial nature of the task. The novelty and uniqueness scores are consistently high. For these tasks, negative $\alpha$ appears to help construct increase the validity scores.

Figure 16 shows qualitatively very similar results for fixed entropy distributions.

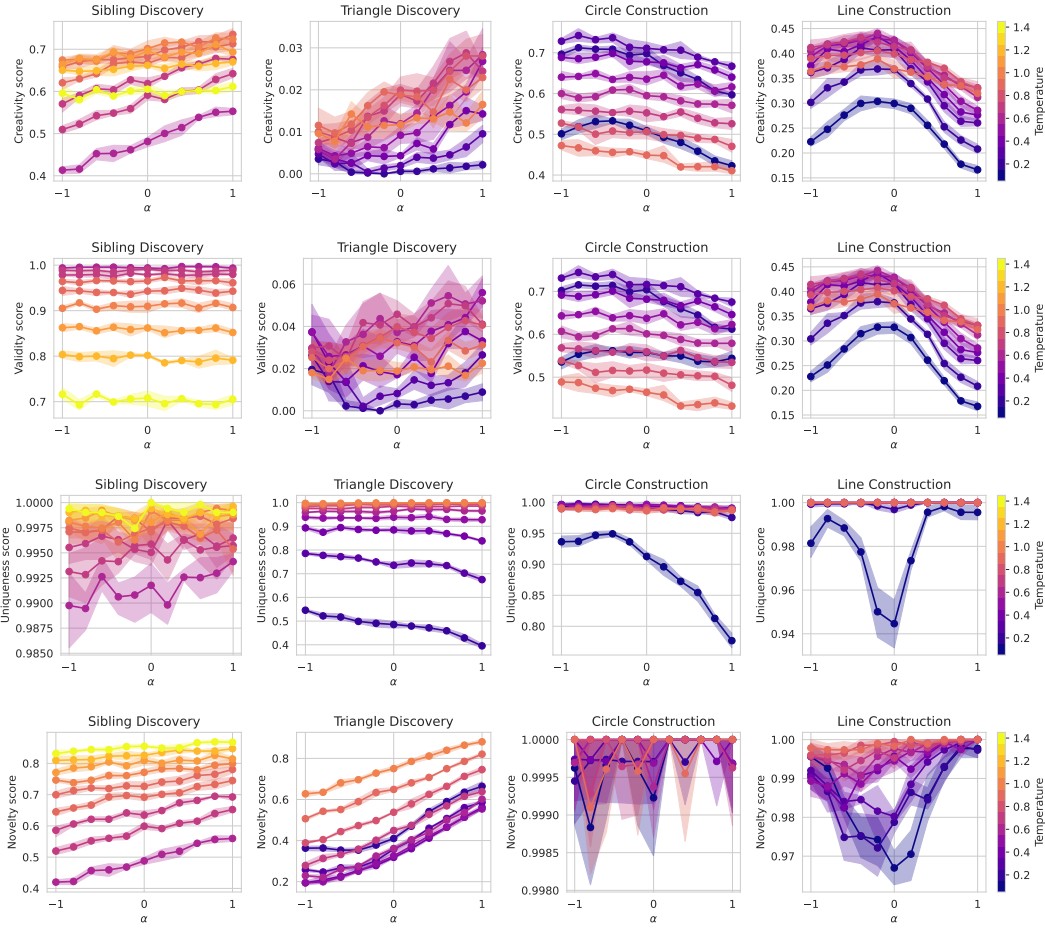

Figure 15: Breakdown of the creativity scores into validity, uniqueness, and novelty. Positive $\alpha$ can improve novelty, negative $\alpha$ can improve validity. Results for geodesic distributions $s_\alpha$.

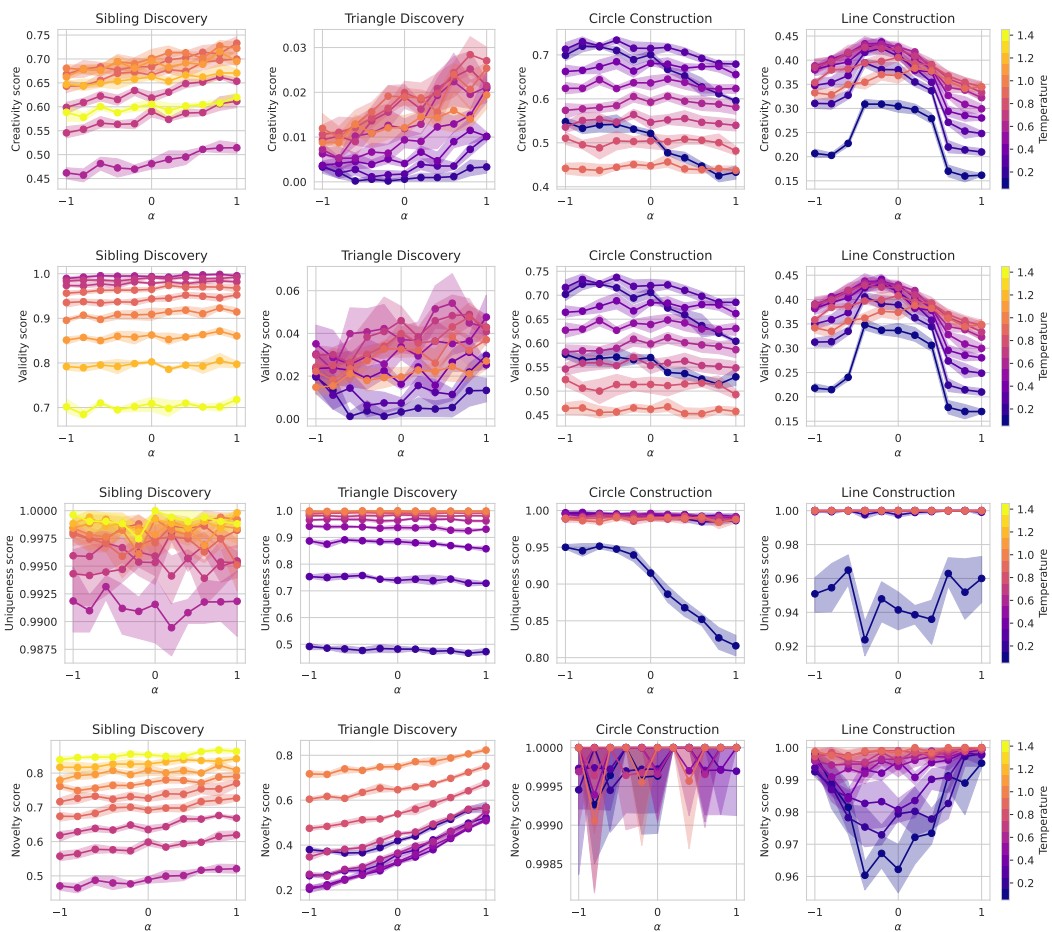

Figure 16: Breakdown of the creativity scores into validity, uniqueness, and novelty. Positive $\alpha$ can improve novelty, negative $\alpha$ can improve validity. Results for fixed entropy distributions $\hat{s}_\alpha$.

## C.3 DIVERGENCE STEERING AND CREATIVE WRITING

Here we report detailed results for different divergence steering settings in the creative writing benchmark (Paech, 2025). For each temperature-$\alpha$ configuration, MiMo generates 96 stories following a range of pre-defined writing prompts. An LLM judge writes an assessment and rates each story according to 21 different criteria, including the aggregate "Overall Impression", which serves as the main score. Figures 17 and 18 show the relationship between temperature, $\alpha$ and the different scores. For many criteria, including the overall impression, there is a connection between good scores and slightly negative $\alpha$. This means that steering away from the "obvious" tokens predicted by the MTP module can lead to better writing performance. On the other hand, some negative criteria connected with excess complexity, such as "Overwrought" and "Purple Prose" tend to increase with negative $\alpha$.

To establish statistical significance, we perform a quadratic regression analysis on the Overall Impression scores. We find that the optimal $\alpha$ is negative and reject the null hypothesis (that $\alpha = 0$ works best) with $p < 0.01$.

## DECLARATION OF LLM USAGE

We used Gemini 2.5 and 3.0 to proofread and occasionally to improve sentence flow and wording.

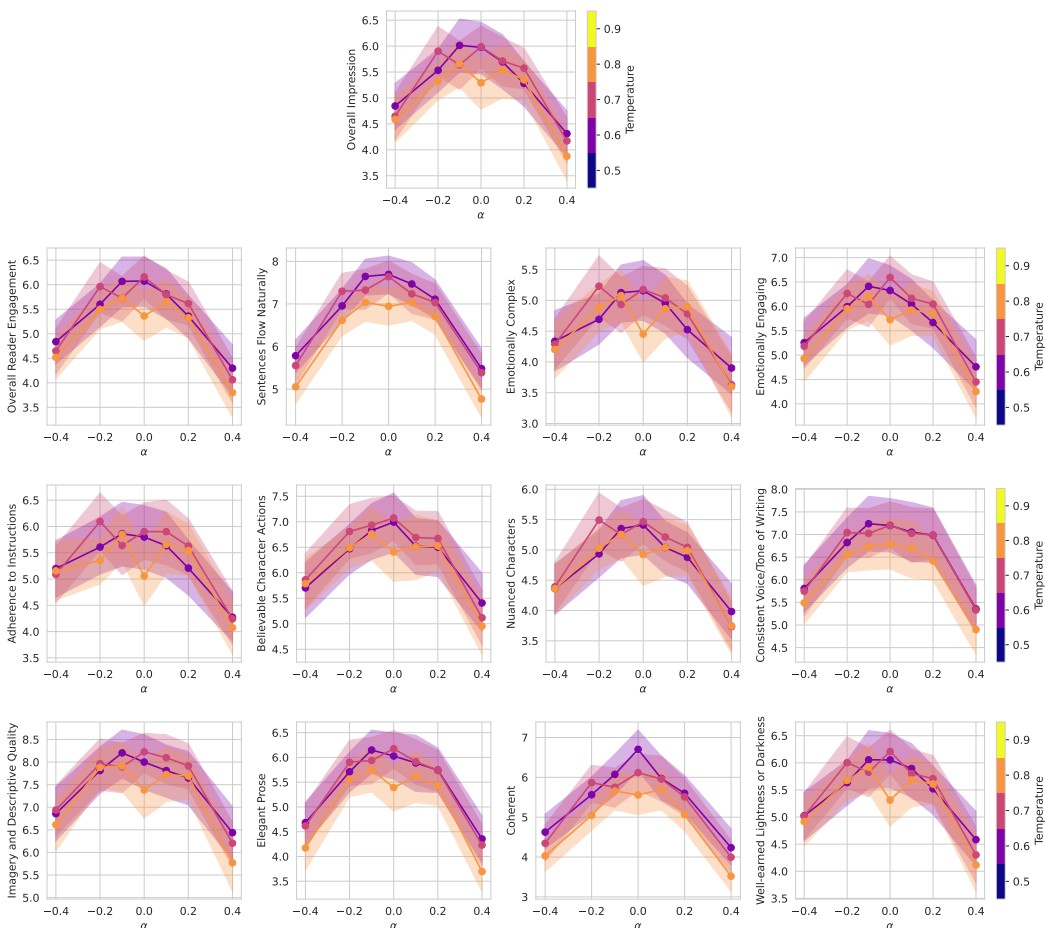

Figure 17: Creative writing scores for positive criteria, for different values of temperature and $\alpha$. Higher value means better. At the top, the aggregate 'Overall Impression' is shown. For many criteria, including Overall Impression, we observe the highest scores for $\alpha = -0.1$.

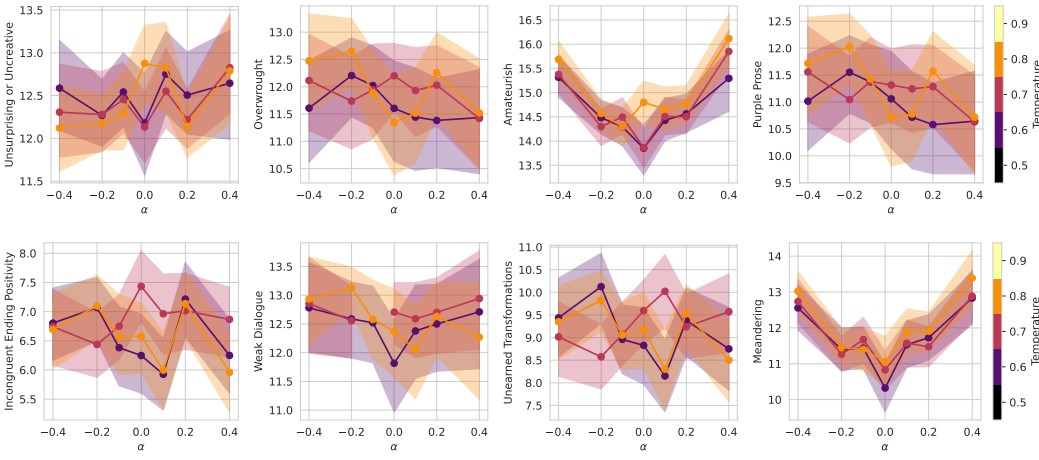

Figure 18: Creative writing scores for negative criteria (lower means better).

