chose this model for two reasons. First, as a modern, high-quality 7B parameter model, its base pre-training incorporates an MTP objective, providing the built-in auxiliary prediction heads necessary for calculating the MTD without any post-hoc modification. Second, its compact size, comparable to Llama 3 8B (Dubey et al., 2024), allows for the efficient, large-scale experimentation required to statistically validate our hypotheses across diverse tasks. We note that while MiMo-7B was trained with an MTP objective from the outset, a similar setup could be achieved for other models by keeping the base model frozen and training an MTP head using standard teacher-student distillation (Schmidhuber, 1992; Hinton et al., 2015) with a fraction of the original data and compute.