# OpenReview forum: "Multiple Token Divergence: Measuring and Steering In-Context Computation Density"
_ICLR.cc/2026/Conference — ICLR 2026 Poster_

### Official Review · Reviewer_Rw1u · 2025-10-15

**Soundness:** 3
**Presentation:** 3
**Contribution:** 3
**Rating:** 6
**Confidence:** 3

**Summary:**

This paper introduces Multiple Token Divergence (MTD), a novel and lightweight metric for measuring the in-context computational effort of language models. MTD is defined as the KL divergence between the output distribution of the full model and that of a shallow, auxiliary prediction head. The core intuition is that a large divergence implies the full model is performing complex computations that cannot be approximated by a simple shortcut. The authors demonstrate that MTD can be computed from existing pre-trained models with Multiple Token Prediction (MTP) heads without requiring any new training. They also propose Divergence Steering, a decoding method that uses MTD to control the "computational character" of generated text by interpolating between the full and shallow output distributions. Empirical results show MTD is a more effective measure of task complexity than prior methods like Prediction of Hidden States (PHi). On mathematical reasoning, MTD correlates positively with problem difficulty, and lower MTD values are associated with more accurate reasoning.

**Strengths:**

**Simplicity and Practicality:** The primary strength of MTD is its simplicity compared to prior methods like PHi. It avoids invasive architectural changes, complex loss functions, and unstable training. The ability to compute it post-hoc on models already equipped with MTP heads (like MiMo-7B) makes it a highly practical tool for analysis.

**Novel Decoding Method:** Divergence Steering is a genuinely new mechanism for controlling generation. It introduces a steering parameter, α, that is conceptually orthogonal to temperature—controlling "computational character" rather than just entropy. The use of geodesic interpolation for this process is principled and elegant.

**Strong Empirical Analysis:** The paper provides a robust empirical validation of MTD as a metric. The head-to-head comparison with PHi on synthetic tasks clearly demonstrates that MTD (with latest embedding access) provides a cleaner signal of computational complexity . The findings on the MATH dataset—that MTD correlates positively with difficulty while NLL correlates negatively—are insightful and effectively decouple the concepts of model surprise and computational effort.

**Isolating Computational Effort:** The paper makes a key contribution by showing that allowing the auxiliary head access to the latest token embedding enables the MTD metric to specifically isolate information gain attributable to non-trivial computation, rather than just novel information in the current token.

**Weaknesses:**

**Limited Impact of Steering on Reasoning:** The most significant weakness is the acknowledged failure of Divergence Steering to improve performance on the core reasoning tasks analyzed. While the method shows interesting, task-dependent effects on toy creative problems, its inability to enhance mathematical reasoning in pre-trained models severely limits the practical impact of the paper's second main contribution.

**Ambiguous Interpretation of the MTD Signal:** The paper finds that lower MTD is associated with correct reasoning on MATH and GSM-8k . This is in direct contrast to prior work where higher PHi loss was linked to correctness. The authors' hypothesis that this is model-dependent is plausible but leaves the interpretation of the MTD signal ambiguous. It is unclear whether high MTD should be interpreted as "deep, productive thinking" or "inefficient, confused thrashing." This ambiguity complicates its use as a straightforward optimization target.

**Conditional "No-Training" Claim:** The claim that MTD requires no additional training is conditional on using a model that already has an MTP head. Most widely-used open-source models do not. The proposed workaround—training a head via distillation —reintroduces the training complexity and cost that MTD is positioned as an alternative to, weakening one of its main selling points.

**Post-Hoc Steering Direction:** In the creative tasks, the optimal steering direction (positive vs. negative α) depends on whether the task is "discovery" or "construction". This is an interesting finding, but the paper offers no method to determine the correct direction for a new, arbitrary task a priori, making the steering method difficult to apply in practice.

**Questions:**

1. The failure of Divergence Steering on reasoning tasks is a critical point. Could you elaborate on your hypothesis for why this occurs? Does it suggest that post-training (e.g., SFT, RLHF) makes a model's generation process too brittle for this kind of decoding intervention, or is MTD fundamentally measuring a property that is not directly aligned with generating a correct reasoning path?

2. Your finding that lower MTD correlates with correctness is intriguing and contrasts with prior PHi results. Beyond model-dependency, could it be that MTD is capturing a different aspect of computation, such as "processing efficiency" or "convergence on a solution," where lower divergence indicates a more confident, streamlined path?

3. For models without a pre-existing MTP head, what is the practical cost (in data and compute) of training a new head via distillation? How does this compare to the cost of implementing and training a full PHi model, and does it maintain a significant advantage in terms of simplicity and stability?

4. Given that the optimal steering direction (α) is task-dependent, how might a practitioner apply Divergence Steering to a novel task where this direction is unknown? Have you explored any methods for dynamically adapting α during generation?

---

> ### Author Response · Authors · 2025-11-27
>
> Thank you for the thorough assessment and for highlighting the practical implications of our work. We have addressed your concerns regarding the cost of MTD and the utility of Divergence Steering below.
>
> > The failure of Divergence Steering on reasoning tasks is a critical point. Could you elaborate on your hypothesis for why this occurs? Does it suggest that post-training (e.g., SFT, RLHF) makes a model's generation process too brittle for this kind of decoding intervention, or is MTD fundamentally measuring a property that is not directly aligned with generating a correct reasoning path?
>
> You asked if the failure of steering on reasoning suggests that post-training makes the model brittle, or if MTD is misaligned with correctness. While both answer are likely true to some extent, there is some interesting nuance to the latter. Our results suggest that there could be a distinction between convergent and divergent tasks:
> - **Convergent (Reasoning):** As our correlation results show, correct reasoning is "efficient"—it follows a low-MTD path. Steering away from this (increasing MTD) disrupts the convergence on the correct answer.
> - **Divergent (Creativity):** Creative tasks benefit from exploring non-obvious paths. We have added a new experiment on the Creative Writing Benchmark (see General Response). There, we found that steering towards higher computational effort (negative α) significantly improves the quality of the output (p<0.01). This makes Divergence Steering a useful tool for enhancing _divergent_ thinking, perhaps similar to how high temperature is used for brainstorming but not for math.
>
> > **Ambiguous Interpretation of the MTD Signal:** The paper finds that lower MTD is associated with correct reasoning on MATH and GSM-8k . This is in direct contrast to prior work where higher PHi loss was linked to correctness. The authors' hypothesis that this is model-dependent is plausible but leaves the interpretation of the MTD signal ambiguous. It is unclear whether high MTD should be interpreted as "deep, productive thinking" or "inefficient, confused thrashing." This ambiguity complicates its use as a straightforward optimization target.
>
> We agree that the MTD signal can be ambiguous. It measures computational effort, which can be used for both "deep, productive thinking" and "inefficient, confused thrashing". There is most likely no bullet-proof way of telling which one it is in any given case. We still believe that it is an important metric. Our experiments suggest that, at least for the models and problems we tested, *low* MTD is a sign of efficient and correct reasoning. However, the results for Divergence Steering, including the new Creative Writing results, show that for certain tasks, the less obvious and predictable path can be better. Perhaps this again represents a rough analogue to convergent vs. divergent thinking.
>
> > **Conditional "No-Training" Claim:** ...
>
> > For models without a pre-existing MTP head, what is the practical cost (in data and compute) of training a new head via distillation? How does this compare to the cost of implementing and training a full PHi model, and does it maintain a significant advantage in terms of simplicity and stability?
>
> You raise a valid concern that training an MTP head via distillation reintroduces complexity. We addressed this with new experiments using Mistral 7B (see General Response).
> **Low Cost:** We found that training a functional MTP head takes only ~10 hours on a single consumer GPU (RTX 4090).
> **Simplicity:** Unlike PHi, which requires tuning a noisy information bottleneck (often unstable), MTP training is standard supervised learning. This confirms that even for models without pre-existing heads, MTD remains a low-cost, stable alternative to PHi.

---

> ### Author Response · Authors · 2025-11-27
>
> > **Post-Hoc Steering Direction:** ...
>
> >Given that the optimal steering direction (α) is task-dependent, how might a practitioner apply Divergence Steering to a novel task where this direction is unknown? Have you explored any methods for dynamically adapting α during generation?
>
> We view α similarly to temperature or top-k: a hyperparameter that governs the "mode" of generation. Based on our experiments (Discovery vs. Construction in the algorithmic toy tasks, and now Creative Writing), we propose the following heuristic:
> - **Positive α:** Use when the goal is simple precision, straightforwardness, or preventing overfitting.
> - **Negative α:** Use when the goal is structural novelty, non-triviality, or "anti-speculation" (e.g., creative writing, adherence to complex constraints).
>
> While we have not yet explored dynamic adaptation during generation, we believe these heuristics provide a practical starting point for practitioners.
>
> > Your finding that lower MTD correlates with correctness is intriguing and contrasts with prior PHi results. Beyond model-dependency, could it be that MTD is capturing a different aspect of computation, such as "processing efficiency" or "convergence on a solution," where lower divergence indicates a more confident, streamlined path?
>
> Yes, as we describe in section 3, MTD and PHi are not measuring exactly the same thing. The correct interpretation of MTD depends, among other things, on the model capability: If the model is so good that even the MTP module can predict the correct steps almost all of the time, then having high MTD would mean that the model is diverging from the these streamlined but adequate solutions, potentially overthinking things and making it worse. However, if the problem is difficult for the model, then efficiency and streamlining can become oversimplification.
>
> We believe our new results fundamentally address the primary weaknesses you identified. The Creative Writing experiment proves that Divergence Steering has practical utility for divergent tasks, converting the "failure" on reasoning into a clear distinction of use-cases. Furthermore, the ~10-hour training time for Mistral MTP heads confirms that MTD is a low-cost, accessible alternative to PHi. In light of this new evidence for the method's utility and practicality, we hope you will consider raising your score.

---

### Official Review · Reviewer_RBiN · 2025-10-24

**Soundness:** 3
**Presentation:** 3
**Contribution:** 3
**Rating:** 6
**Confidence:** 3

**Summary:**

This paper introduces Multiple Token Divergence (MTD), a metric designed to measure the in-context computational effort of a language model at each token. The authors propose that standard metrics like next-token loss fail to capture the complexity of the reasoning required to produce a token. Their method addresses the shortcomings of prior work (like the PHi loss).

MTD is defined as the KL divergence between the full model's next-token output distribution and the distribution of a shallow, auxiliary prediction head (an MTP module). The core intuition is that if a "shortcut" shallow model can easily predict the same token as the full, deep model, the full model isn't performing complex computation. A large divergence, however, implies the deeper layers are performing non-trivial work. A key refinement is to provide the shallow MTP module with access to the current token embedding, which helps MTD isolate computational effort from the simple information gain of the new token.

The authors also introduce Divergence Steering, a decoding technique that uses the MTD signal to control the "computational character" of the generated text by interpolating between the full and shallow models' distributions. This can bias generation to be more "anti-speculative" (favoring computationally intensive tokens).

**Strengths:**

- The paper's most fascinating result (sec 4.2) is the decoupling of computational effort (MTD) from predictive plausibility (NLL). The finding that MTD correlates positively with MATH problem difficulty while NLL correlates negatively is a significant contribution.
- The discovery that MTD and NLL are anti-correlated with respect to problem difficulty is very interesting. It provides a new, orthogonal axis for analyzing model behavior. NLL measures "plausibility" or "surprise," while MTD measures "effort".
- The use of geodesic interpolation (via the Fisher-Rao metric) to navigate between the two output distributions ($\pi$ and $\pi_{MTP}$) is a principled and technically sophisticated approach, avoiding the pitfalls of a naive linear or log-linear interpolation.

**Weaknesses:**

- The paper's discussion briefly notes that MTD may "entangle genuine computational effort with memorization". This is a problem because the shallow MTP head is likely trained to be very good at predicting common, high-frequency (like, memorized) n-grams. A high MTD might simply signal that the full model is generating a novel or rare sequence (for example, a specific fact or a unique turn of phrase) that the shallow head couldn't possibly predict, which is not necessarily the same as in-context computation.
- In section 4.2. MTD is explained as a measure of "sophisticated in-context computation", but the experiments find that lower MTD correlates with correct reasoning. This seems contradictory: if high MTD means high computational effort, but low MTD means correctness, it suggests that high MTD might actually be a signal of the model "struggling" or engaging in inefficient/failed computation, rather than successful, deep reasoning.
- The MTD metric is strongly relative, because it measures the divergence between a full model and a specific shallow MTP head. The paper's results are contingent on the architecture of the MTP head used. If the MTP head were slightly more powerful or weaker, how would the MTD values and their correlations change?
- The main LLM experiments are carried out on a single 7B-parameter model (MiMo-7B). While this model is well-suited for the study, it remains unclear if these findings (especially the crucial MTD-difficulty and MTD-correctness correlations) generalize to other model architectures.
-

**Questions:**

The finding that lower MTD correlates with correct reasoning (sec 4.2, fig 7c) is one of the most interesting results of the paper. It seems to contrast with the initial motivation of MTD as a measure of "sophisticated" or "non-trivial" computation, which you might associate with successful reasoning.
Could you elaborate on this relationship? Does this finding imply that high MTD is a signal of inefficient or failed computation (like, the model is "struggling" or "thrashing" on its way to a wrong answer), rather than a signal of successful, deep reasoning? How do you handle this with the fact that high MTD also correlates with overall problem difficulty?

---

> ### Author Response · Authors · 2025-11-27
>
> Thank you for the insightful comments. We address your specific points below.
>
> > The paper's discussion briefly notes that MTD may "entangle genuine computational effort with memorization". This is a problem because the shallow MTP head is likely trained to be very good at predicting common, high-frequency (like, memorized) n-grams. A high MTD might simply signal that the full model is generating a novel or rare sequence (for example, a specific fact or a unique turn of phrase) that the shallow head couldn't possibly predict, which is not necessarily the same as in-context computation.
>
> We agree that distinguishing between memorization and computation is a fundamental challenge in interpreting LLMs. As we note in the paper, giving the MTP module access to the latest embedding helps mitigate this by allowing "trivial" pattern matching to be handled by the shallow head, leaving MTD to capture more complex dependencies. However, we acknowledge that MTD is one signal within a larger, necessary research program to disentangle these processes.
>
> > In section 4.2. MTD is explained as a measure of "sophisticated in-context computation", but the experiments find that lower MTD correlates with correct reasoning. This seems contradictory: if high MTD means high computational effort, but low MTD means correctness, it suggests that high MTD might actually be a signal of the model "struggling" or engaging in inefficient/failed computation, rather than successful, deep reasoning.
> >
> > The finding that lower MTD correlates with correct reasoning (sec 4.2, fig 7c) is one of the most interesting results of the paper. It seems to contrast with the initial motivation of MTD as a measure of "sophisticated" or "non-trivial" computation, which you might associate with successful reasoning. Could you elaborate on this relationship? Does this finding imply that high MTD is a signal of inefficient or failed computation (like, the model is "struggling" or "thrashing" on its way to a wrong answer), rather than a signal of successful, deep reasoning? How do you handle this with the fact that high MTD also correlates with overall problem difficulty?
>
> High MTD indicates high computational exertion. While difficult problems generally demand more effort (positive correlation with difficulty), _within_ a specific problem attempt, excessive exertion can signal "thrashing" or a lack of convergence on a clean solution. Our results suggest that, at least for the kinds of problems and models we evaluated, correct reasoning tends to be efficient (lower MTD compared to incorrect reasoning on the same problem). On the other hand, our new Creative Writing results (see General Response) show that for open-ended tasks, "struggling" or steering towards "anti-speculative" tokens (negative α) actually improves quality. Perhaps one way to interpret these results is that high MTD is undesirable for convergent reasoning, but can be beneficial for divergent creativity.
>
> > The MTD metric is strongly relative, because it measures the divergence between a full model and a specific shallow MTP head. The paper's results are contingent on the architecture of the MTP head used. If the MTP head were slightly more powerful or weaker, how would the MTD values and their correlations change?
>
> We have conducted new experiments to answer this question directly (see General Response). We trained MTP modules of three different capacities (Small, Medium, Large) for the Mistral 7B model.
> While the absolute MTD values decrease as the MTP head becomes stronger (as expected), the correlations with task difficulty and correctness remain stable across all sizes. This indicates that MTD is a robust relative metric, not highly sensitive to the exact architecture and capacity of the auxiliary head.
>
> > The main LLM experiments are carried out on a single 7B-parameter model (MiMo-7B). While this model is well-suited for the study, it remains unclear if these findings (especially the crucial MTD-difficulty and MTD-correctness correlations) generalize to other model architectures.
>
> Our new experiments with Mistral 7B (see General Response) confirm that our findings generalize to standard pre-trained models where the MTP head is trained post-hoc.
> We observed the similar positive correlation between MTD and difficulty, and the similar predictive power for correctness. While the correlations are slightly weaker than for MiMo (likely because MiMo was trained _end-to-end_ with the MTP objective), they remain statistically significant and directionally consistent.
>
> We believe these clarifications and new results resolve the weaknesses you highlighted, and we would appreciate it if you could reconsider your score.

---

### Official Review · Reviewer_5KTN · 2025-10-31

**Soundness:** 3
**Presentation:** 3
**Contribution:** 3
**Rating:** 6
**Confidence:** 3

**Summary:**

This paper introduces Multiple Token Divergence (MTD), an alternative to PHi for measuring computational effort by computing the KL divergence between a model's output distribution and that of a shallow auxiliary prediction head. Based on this, the authors propose Divergence Steering, a decoding method that uses MTD to control the "computational character" of generated text.

**Strengths:**

1. The paper is clear and well-motivated: the paper addresses limitations of PHi with a simpler, non-invasive method.
2. The evaluation is fairly comprehensive, covering both pre-trained models as well as those trained from scratch. While the model may require some adaptation (if it doesn't have an MTP head), it's less invasive than PHi.
3. MTD shows good correlation with task complexity and difficulty (e.g., on MATH). Furthermore, it's effective for CoT rationale selection when combined with NLL, giving MTD practical use.

**Weaknesses:**

1. The paper relies heavily on informal notions of concepts like "complexity", as well as "interesting" and "boring" tasks, instead of formal definitions.
2. The authors only focus on one "interesting" task in Section 3.1. It would be interesting to see if these results generalize to other "interesting" tasks.
3. Section 4.2 is missing a direction comparison to PHi for pre-trained models. Looking at the PHi paper, they report lower correlation with reasoning difficulty, but this could be due to other confounding factors (e.g., different pre-trained model). A direct comparison would be valuable here.
4. Likewise, more work is needed to understand the role of Divergence Steering with pre-trained models. Without such analysis, the impact of Divergence Steering is diminished.

**Questions:**

See weaknesses above.

---

> ### Author Response · Authors · 2025-11-27
>
> Thank you the positive assessment of our motivation and clarity, and for the constructive feedback regarding the definitions and the breadth of our evaluation. We have addressed these points below and included new experiments to strengthen the paper.
>
> > The paper relies heavily on informal notions of concepts like "complexity", as well as "interesting" and "boring" tasks, instead of formal definitions.
>
> We agree that formalizing these concepts is crucial. Our work is grounded in the Minimum Description Length (MDL) principle: a task is "interesting" or "complex" if the most compact description of the solution program (given the training data) is long.
>
> **"Boring" tasks:** In our experiments, tasks like memorization, random sequences, or copying correspond to trivial programs (e.g., simple lookup or "output uniform"), which have low description length.
> **"Interesting" tasks:** The In-Context Language Learning (ICLL) task requires synthesizing a Probabilistic Finite Automaton (PFA) structure, which has a high description length. Similarly, for the MATH dataset, the "complexity" corresponds to the length and depth of the reasoning chain required.
>
> We added a clarification of the mentioned terms to the introduction.
>
> > The authors only focus on one "interesting" task in Section 3.1. It would be interesting to see if these results generalize to other "interesting" tasks.
>
> Section 4.1 serves as a controlled environment to validate the metric against ground-truth complexity (PFA size). However, we believe our evaluation now spans three distinct types of "interesting" tasks:
> 1. **Algorithmic Induction:** In-Context Language Learning (Section 4.1).
> 2. **Mathematical Reasoning:** The MATH dataset (Section 4.2), where we show MTD correlates with problem difficulty.
> 3. **Creative Generation:** As detailed in our General Response, we have added the Creative Writing Benchmark, where the task is to generate high-quality, open-ended stories.
>
> We believe this covers a robust spectrum of "interesting" behaviors (Pattern Induction, Reasoning, and Creativity).
>
> > Section 4.2 is missing a direction comparison to PHi for pre-trained models. Looking at the PHi paper, they report lower correlation with reasoning difficulty, but this could be due to other confounding factors (e.g., different pre-trained model). A direct comparison would be valuable here.
>
> A direct comparison on pre-trained models is difficult because PHi is architecturally invasive. PHi requires inserting a bottleneck layer _between_ existing layers and training it, which significantly alters the pre-trained model's internal dynamics and can invalidate the pre-trained weights (unless extensive fine-tuning is performed). In contrast, a key advantage of MTD is that it is non-invasive and can be applied post-hoc. However, to address the underlying concern—validating MTD on standard pre-trained architectures—we have conducted experiments on Mistral 7B (see General Response). The results confirm that the MTD correlations hold for standard pre-trained models, even when the MTP head is trained post-hoc via distillation.
>
> > Likewise, more work is needed to understand the role of Divergence Steering with pre-trained models. Without such analysis, the impact of Divergence Steering is diminished.
>
> We agree, and we have addressed this with the new Creative Writing experiment (see General Response). We applied Divergence Steering to the MiMo model on the Creative Writing Benchmark and found a statistically significant relationship (p<0.01) where a negative steering parameter maximizes the "Overall Impression" score, as well as the majority of all other scores. This demonstrates that Divergence Steering generalizes effectively to pre-trained models in creative domains, actively helping the model avoid "boring" or trivial tokens.
>
> We believe the new Mistral 7B experiments directly address your concern regarding the comparison to pre-trained models. The inclusion of the Creative Writing Benchmark further demonstrates that our methods generalize to other "interesting" domains beyond the single task in Section 4.1, and shows the practical value of Divergence Steering. As we have now provided the broader empirical validation, we would appreciate if you reconsider your assessment and confidence in the paper's contribution.

---

### Author Response · Authors · 2025-11-27
**Summary of New Results**

*We have restructured and improved the general responses in light of the updated rebuttal process.*

We thank the reviewers for their comments and suggestions. Common critiques centered on two main issues: **(1)** The reliance on the MiMo 7B model (trained from scratch with MTP) and the lack of empirical support for the "post-hoc" application of MTD to standard LLMs; and **(2)** the limited demonstrated utility of Divergence Steering beyond toy tasks. They led us to add two major experimental results, which we believe greatly strengthen our empirical corroboration of MTD and Divergence Steering. The two new results are:

**The important characteristics of the MTD (positive correlation with task difficulty, low MTD being predictive of correct reasoning) remain robust across model architectures and MTP module capacities.**  We train dedicated MTP modules with different capacities post-hoc for the Mistral 7B model. This can be done in under ten hours on a single RTX 4090 GPU. All the main experimental findings from the MiMo model are replicated. This directly addresses weaknesses 3&4 from reviewer RBiN (*"...If the MTP head were slightly more powerful or weaker, how would the MTD values and their correlations change?", "...it remains unclear if these findings (especially the crucial MTD-difficulty and MTD-correctness correlations) generalize to other model architectures."*), and weakness 3 (*"...The proposed workaround—training a head via distillation —reintroduces the training complexity and cost that MTD is positioned as an alternative to..."*)  as well as question 2 from reviewer Rw1u (*"For models without a pre-existing MTP head, what is the practical cost (in data and compute) of training a new head via distillation?..."*).

**Divergence Steering helps the creative writing abilities of the pre-trained MiMo model.** We test generation with Divergence Steering on the creative writing benchmark and find that negative $\alpha$ improves the scores. This result addresses weakness 4 from reviewer 5KTN (*"...more work is needed to understand the role of Divergence Steering with pre-trained models..."*), and weakness 1 from reviewer Rw1u (*"The most significant weakness is the acknowledged failure of Divergence Steering to improve performance on the core reasoning tasks analyzed..."*).

These new results are detailed below and in the updated paper, where all changes are clearly marked (written in green). Other concerns are addressed in the responses to the individual reviews.

---

> ### Author Response · Authors · 2025-12-02
> **Result: Generalization to Mistral 7B and Different MTP Capacities**
>
> To demonstrate that MTD is a practical metric for standard open-source models, we trained MTP modules of varying capacities for the Mistral 7B v0.3 model. As shown in the table below, training a functional MTP module requires only ~10 hours on a single consumer GPU (RTX 4090). This validates our claim that MTD is a non-invasive, low-cost tool for analysis, even for models without native MTP heads. The relationship between MTP capacity and loss is consistent: larger modules reduce MTD.
>
> *Comparison between MiMo and Mistral models with different MTP capacities:*
>
> | Metric  | MiMo   | Mistral MTP Small | Mistral MTP Medium | Mistral MTP Large |
> | -- | -- | -- | - | - |
> | Vocabulary Size | 151936 | 32000             | 32000              | 32000             |
> | NLL             | 1.345  | 0.943  | 0.943              | 0.943             |
> | MTP Loss        | 1.425  | 1.298             | 1.217              | 1.178             |
> | MTD             | 0.469  | 0.553             | 0.476              | 0.429             |
> | RTX 4090 hours  | —      | 9.38              | 10.25              | 11.08             |
>
>
> Our core findings regarding the nature of MTD hold true for Mistral, robustly across different MTP sizes. MTD consistently correlates positively with problem difficulty (MATH dataset), whereas NLL correlates negatively. This confirms that MTD captures "computational effort" distinct from "surprise".
>
> *Partial correlation of MTD and NLL with difficulty:*
> | Correlation Type   | MiMo   | Mistral MTP Small  | Mistral MTP Medium  | Mistral MTP Large |
> |--|--|--|--|--|
> | Provided Solutions: MTD-Diff (ctrl NLL) | 0.162 [0.137, 0.189]    | 0.109 [0.079, 0.138]    | 0.094 [0.065, 0.122]    | 0.096 [0.066, 0.128]    |
> | Provided Solutions: NLL-Diff (ctrl MTD)  | -0.239 [-0.266, -0.212] | -0.153 [-0.180, -0.127] | -0.143 [-0.172, -0.115] | -0.144 [-0.173, -0.115] |
> | Generated CoT Solutions: MTD-Diff (ctrl NLL) | 0.199 [0.189, 0.208]    | 0.050 [0.041, 0.058]    | 0.044 [0.035, 0.053]    | 0.043 [0.034, 0.052]    |
> | Generated CoT Solutions: NLL-Diff (ctrl MTD) | -0.205 [-0.215, -0.195] | -0.041 [-0.050, -0.033] | -0.047 [-0.056, -0.038] | -0.047 [-0.056, -0.038] |
>
>
> Lower MTD is consistently predictive of correct reasoning. As shown in the table below, selecting CoTs based on lower MTD (or both lower NLL and MTD) significantly outperforms random selection.
>
> *Probability of selecting the correct answer from a (Correct, Incorrect) pair:*
> | Criterion  |  MiMo  |  Mistral MTP Small  |  Mistral MTP Medium |  Mistral MTP Large  |
> |-|:--:|:--:|:---:|:--:|
> | MATH, lower NLL    | 73.3%, [71.9, 74.8] | 59.1%, [57.6, 60.6] | 59.1%, [57.6, 60.6] | 59.1%, [57.6, 60.6] |
> | MATH, lower MTD    | 67.1%, [65.5, 68.5] | 57.2%, [55.6, 58.6] | 57.5%, [56.0, 59.0] | 57.8%, [56.2, 59.3] |
> | MATH, both lower   | 80.4%, [78.5, 81.9] | 63.0%, [61.1, 64.9] | 62.7%, [60.8, 64.6] | 62.7%, [60.9, 64.6] |
> | GSM-8k, lower NLL  | 72.2%, [69.1, 75.2] | 64.9%, [63.1, 66.7] | 64.9%, [63.1, 66.7] | 64.9%, [63.1, 66.7] |
> | GSM-8k, lower MTD  | 66.0%, [62.4, 69.3] | 59.2%, [57.5, 61.0] | 61.8%, [60.1, 63.5] | 61.3%, [59.6, 63.1] |
> | GSM-8k, both lower | 75.5%, [71.7, 79.2] | 69.9%, [67.6, 72.1] | 71.4%, [69.3, 73.5] | 70.9%, [68.8, 73.0] |
>
>
> For more details, please see Appendix B.2 and C.1 in the updated paper.

---

> > ### Author Response · Authors · 2025-12-02
> > **Result: Divergence Steering Improves Creative Writing**
> >
> > To address the utility of Divergence Steering, we evaluated the method on the [Creative Writing Benchmark](https://github.com/EQ-bench/creative-writing-bench). We generated 96 stories per configuration using MiMo and evaluated them according to various criteria using an LLM judge (Claude Sonnet 4.5). We find that steering away from simple MTP predictions (negative α) yields the best performance. The table below shows that the majority of criteria, including the main aggregate grade 'Overall Impression', have the best scores (highest for positive attributes, lowest for negative ones marked with $\downarrow$) for negative $\alpha$, i.e. when steering away from the easily predictable tokens.
> >
> > *Creative writing criteria which reach the best score for a given $\alpha$ value:*
> >
> > | $\alpha=-0.4$ | $\alpha=-0.2$                           | $\alpha=-0.1$                     | $\alpha=0.0$                     | $\alpha=0.1$                               | $\alpha=0.2$ | $\alpha=0.4$              |
> > | ------------- | --------------------------------------- | --------------------------------- | -------------------------------- | ------------------------------------------ | ------------ | ------------------------- |
> > | —             | Unsurprising or Uncreative $\downarrow$ | **Overall Impression**            | Overall Reader Engagement        | Unearned Transformations $\downarrow$      | —            | Overwrought $\downarrow$  |
> > |               |                                         | Emotionally Complex               | Consistent Voice/Tone of Writing | Incongruent Ending Positivity $\downarrow$ |              | Purple Prose $\downarrow$ |
> > |               |                                         | Emotionally Engaging              | Coherent                         |                                            |              |                           |
> > |               |                                         | Adherence to Instructions         | Sentences Flow Naturally         |                                            |              |                           |
> > |               |                                         | Believable Character Actions      | Amateurish $\downarrow$          |                                            |              |                           |
> > |               |                                         | Nuanced Characters                | Meandering $\downarrow$          |                                            |              |                           |
> > |               |                                         | Imagery and Descriptive Quality   |                                  |                                            |              |                           |
> > |               |                                         | Elegant Prose                     |                                  |                                            |              |                           |
> > |               |                                         | Well-earned Lightness or Darkness |                                  |                                            |              |                           |
> > |               |                                         | Weak Dialogue $\downarrow$        |                                  |                                            |              |                           |
> >
> > To further establish significance, we perform a quadratic regression analysis and confirm that the optimal $\alpha$ is negative ($\alpha \approx -0.1$) with high statistical significance (p<0.01). This suggests that Divergence Steering can help models avoid "boring" or "obvious" tokens in creative contexts, providing utility beyond toy tasks.
> >
> > More details can be found in Section 4.2, as well as Appendix B.3 and C.3 in the updated paper.

---

### Meta-Review · Area_Chair_9mJ9 · 2026-01-07

**Summary:**

This paper introduces a new lightweight metric, Multiple Token Divergence, which measures the impact of the additional compute available in larger language models relative to a shallow prediction head (such as the Multiple Token Prediction heads used in speculative decoding) on the token output distribution. Specifically, MTD is defined as the KL divergence between the output distribution of the full model and that of an auxiliary prediction head. The key idea is that divergences between the predictions of the larger and smaller model are indicative of additional task complexity captured only by the larger model and not the smaller one. This metric is shown to be positively correlated to task difficulty on the MATH dataset, while standard negative log likelihood loss is negatively correlated. The authors argue that the MTD metric can be used as the basis for a new decoding method, Divergence Steering, which interpolates between the two distributions with geodesic interpolation.

The MTD method is established as a clear improvement on the previous Prediction of Hidden States (PHi) approach. The main weakness, also highlighted by reviewers, is in the difficulty of interpreting the method due to the reliance on comparison to an auxiliary model. (Which applies to the MiMo model, but is especially significant when the MTP head is trained from scratch.) This manifests in the problem of setting the \alpha hyperparameter in Divergence Steering a priori and the failure of this approach to improve performance on the MATH dataset, with NLL being a better heuristic for choosing between correct answers. The additional results showing improvements on creative writing provide a single use case where the steering method is successful, but do not address the more general issue.

Notes:

I think this is a weak accept. The method is sound and logical, and the case for improvement over the prior PHi method is clear. The authors reasonably compare their new hyperparameter to temperature in arguing that there is not a single best value for any use case, but the substantial dependence on the MTP head makes their method less effective and much harder to use.

**Reviewer Concerns:**

see "Summary" text

**Reviewer Scores:**

see "Summary" text

---

### Decision · Program_Chairs · 2026-01-26

Accept (Poster)